# Conformal Meta-learners for Predictive Inference of Individual Treatment Effects

**Ahmed M. Alaa**
UC Berkeley and UCSF
amalaa@berkeley.edu

**Zaid Ahmad**
UC Berkeley
zaidahmad@berkeley.edu

**Mark van der Laan**
UC Berkeley
laan@stat.berkeley.edu

## Abstract

We investigate the problem of machine learning-based (ML) predictive inference on individual treatment effects (ITEs). Previous work has focused primarily on developing ML-based "meta-learners" that can provide point estimates of the conditional average treatment effect (CATE)—these are model-agnostic approaches for combining intermediate nuisance estimates to produce estimates of CATE. In this paper, we develop *conformal meta-learners*, a general framework for issuing predictive intervals for ITEs by applying the standard conformal prediction (CP) procedure on top of CATE meta-learners. We focus on a broad class of meta-learners based on two-stage pseudo-outcome regression and develop a *stochastic ordering* framework to study their validity. We show that inference with conformal meta-learners is marginally valid if their (pseudo-outcome) conformity scores stochastically dominate "oracle" conformity scores evaluated on the unobserved ITEs. Additionally, we prove that commonly used CATE meta-learners, such as the *doubly-robust* learner, satisfy a model- and distribution-free stochastic (or convex) dominance condition, making their conformal inferences valid for practically-relevant levels of target coverage. Whereas existing procedures conduct inference on nuisance parameters (i.e., potential outcomes) via weighted CP [1], conformal meta-learners enable direct inference on the target parameter (ITE). Numerical experiments show that conformal meta-learners provide valid intervals with competitive efficiency while retaining the favorable point estimation properties of CATE meta-learners.

**Code:** https://github.com/AlaaLab/conformal-metalearners

## 1 Introduction

Identifying heterogeneity in the effects of interventions across individuals is a central problem in various fields, including medical, political, and social sciences [2, 3, 4]. In recent years, there has been growing interest in developing machine learning (ML) models to estimate heterogeneous treatment effects using observational or experimental data [5, 6, 7, 8, 9]. However, most of these models only provide *point* estimates of the conditional average treatment effect (CATE), which is a deterministic function that describes the expected treatment effect based on a given individual's covariates. In this paper, we focus on quantifying uncertainty in these estimates, which arises from both errors in the model and the variation of individual treatment effects (ITEs) for individuals with the same covariates. We adopt a *predictive inference* approach to this problem, with the goal of devising valid procedures to issue predictive intervals that cover ITEs on unseen data with a predetermined probability.

Traditionally, predictive inference on ITEs has been conducted through Bayesian methods such as BART [8] and Gaussian processes [9]. These methods can provide interval-valued predictions of ITEs through their induced posterior distributions (e.g., posterior credible intervals). However, Bayesian methods tend to be model-specific and cannot be straightforwardly generalized to modern ML models, e.g., transformer-based architectures used to model visual and textual covariate spaces [10]. More importantly, Bayesian methods generally do not provide guarantees on the frequentist coverage of

37th Conference on Neural Information Processing Systems (NeurIPS 2023).

their credible intervals—achieved (finite-sample) coverage depends on the prior [11]. This paper is motivated by the advent of *conformal prediction* (CP), a frequentist alternative that can be used to conduct model-agnostic, distribution-free valid predictive inference on top of any ML model [12, 13, 14]. Throughout this paper, we will study the validity of CP-based procedures for inference of ITEs.

***What makes CP-based inference of ITEs different from its application to the standard regression (supervised) setup?*** The "fundamental problem of causal inference" is that we never observe counterfactual outcomes [15]. That is, our "label" is the ITE which is a difference between two potential outcomes (treated and untreated) for an individual subject—this label is never observed for any given subject because we only ever observe factual outcomes. This poses two challenges [16]:

**(1) How to handle *covariate shift*?** When treatments are assigned to individuals with probabilities that depend on their covariates, then the distributions of covariates in treated and untreated groups differ. Consequently, the distribution of training data differs from that of the target population.

**(2) How to incorporate *inductive biases*?** Unlike supervised learning wherein we fit a single function using examples of covariates and *observed* targets, models of treatment effects cannot be directly fit to the *unobserved* effects. Thus, estimates of treatment effects comprise intermediate estimates of nuisance parameters. Different approaches for combining nuisance estimates entail different inductive priors on the potential outcomes that affect the accuracy of the resulting ITE estimates.

The literature on ML-based CATE estimation focuses on addressing the two questions above. **Covariate shift** affects the generalization performance of ML models—existing CATE estimation models address this problem using importance weighting [17] or balanced representation learning methods for unsupervised domain adaptation [6, 18, 19]. In [5], the notion of "meta-learners" was coined to describe various model-agnostic approaches to incorporating **inductive biases** and combining nuisance estimates. In [5, 20], it was shown that the choice of the meta-learner influences the CATE estimation rates. While the impact of **(1)** and **(2)** on the generalization performance of CATE estimators has been extensively investigated, their impact on the validity and efficiency of predictive inference methods for ITE is less well-understood. This forms the central focus of our paper.

**Contributions.** We propose a CP procedure for predictive inference of ITEs that jointly addresses **(1)** and **(2)** in an end-to-end fashion. Our proposed inference strategy applies the standard CP procedure on top of a broad class of CATE meta-learners based on two-stage *pseudo-outcome* regression. These meta-learners operate by first estimating pseudo-outcomes, i.e., transformed targets that depend on observable variables only, and then regressing the pseudo-outcomes on covariates to obtain point estimates of CATE. We then construct intervals for ITEs by computing the empirical quantile of conformity scores evaluated on pseudo-outcomes in a held-out calibration set. Conformal meta-learners address **(1)** because the distribution of covariates associated with pseudo-outcomes is the same for training and testing data, and they address **(2)** since the calibration step is decoupled from model architecture, enabling flexible choice of inductive priors and the possibility of re-purposing existing meta-learners and architectures that have been shown to provide accurate estimates of CATE.

Conformal meta-learners inherit the guarantees of CP, i.e., their resulting intervals cover pseudo-outcomes on test data with high probability. However, the original CP guarantees do not immediately translate to guarantees on coverage of ITEs. To this end, we develop a unified *stochastic ordering* framework to study the validity of conformal meta-learners for inference on ITEs. We show that inference with conformal meta-learners is valid if their conformity scores satisfy certain stochastic ordering conditions with respect to "oracle" conformity scores evaluated on unobserved ITEs. We prove that some of the commonly used meta-learners, such as the *doubly-robust* learner [20], satisfy a weaker stochastic (or convex) dominance condition which makes them valid for relevant levels of target coverage. Our numerical experiments show that, with careful choice of the pseudo-outcome transformation, conformal meta-learners inherit both the coverage properties of CP as well as the efficiency and point estimation accuracy of their underlying CATE meta-learners.

## 2 Predictive Inference of Individual Treatment Effects (ITEs)

### 2.1 Problem setup

We consider the standard potential outcomes (PO) framework with a binary treatment ([21, 22]). Let $W \in \{0, 1\}$ be the treatment indicator, $X \in \mathcal{X}$ be the covariates, and $Y \in \mathbb{R}$ be the outcome of interest. For each subject $i$, let $(Y_i(0), Y_i(1))$ be the pair of potential outcomes under $W = 0$ and $W = 1$,

respectively. The fundamental problem of causal inference is that we can only observe the *factual* outcome, i.e., the outcome $Y_i = W_i Y_i(1) + (1 - W_i)Y_i(0)$ determined by $W_i$, but we cannot observe the *counterfactual* $Y_i(1 - W_i)$. For $n$ subjects, we assume that the data generation process

$$(X_i, W_i, Y_i(0), Y_i(1)) \overset{iid}{\sim} P(X, W, Y(0), Y(1)), \ i = 1, \ldots, n, \tag{1}$$

satisfies the following assumptions: *(1) Unconfoundedness:* $(Y(0), Y(1)) \perp W \mid X$, *(2) Consistency:* $Y = Y(W)$, and *(3) Positivity:* $0 < P(W = 1 \mid X = x) < 1$, $\forall x \in \mathcal{X}$. These assumptions are necessary for identifying the causal effects of the treatment from a dataset $\{Z_i = (X_i, W_i, Y_i)\}_{i=1}^{n}$. The causal effect of the treatment on individual $i$, known as the *individual treatment effect* (ITE), is defined as the difference between the two potential outcomes, i.e., $Y_i(1) - Y_i(0)$.

Previous modeling efforts (e.g., [5, 6, 7, 8]) have focused primarily on the (deterministic) *conditional average treatment effect* (CATE), i.e., $\tau(x) \triangleq \mathbb{E}[Y(1) - Y(0) \mid X = x]$. In this paper, we focus on the (random) ITE as the inferential target of interest. That is, our goal is to infer the ITE for a given subject $j$ given their covariate $X_j$ and the observed sample $\{Z_i = (X_i, W_i, Y_i)\}_{i=1}$.

The distribution of the observed variable $Z = (X, W, Y)$ is indexed by the covariate distribution $P_X$, as well as the nuisance functions $\pi(x)$, $\mu_0(x)$ and $\mu_1(x)$ defined as follows:

$$\begin{aligned} \pi(x) &= \mathbb{P}(W = 1 \mid X = x), \\ \mu_w(x) &= \mathbb{E}[Y \mid X = x, W = w], \ w \in \{0, 1\}. \end{aligned} \tag{2}$$

The function $\pi(x)$ is known as the *propensity score* and it captures the treatment mechanism underlying the data generation process. Throughout this paper, we assume that $\pi(x)$ is known, i.e., data is drawn from an experimental study or the treatment assignment mechanism is known.

**Predictive Inference of ITEs.** Given the sample $\{Z_i = (X_i, W_i, Y_i)\}_{i=1}^{n}$, our goal is to infer the ITE for a new individual $n + 1$ with covariate $X_{n+1}$. In particular, we would like to construct a predictive band $\widehat{C}(x)$ that covers the true ITE for new test points with high probability, i.e.,

$$\mathbb{P}(Y_{n+1}(1) - Y_{n+1}(0) \in \widehat{C}(X_{n+1})) \geq 1 - \alpha, \tag{3}$$

for a predetermined target coverage of $1 - \alpha$, with $\alpha \in (0, 1)$, where the probability in (3) accounts for the randomness of the training data $\{Z_i\}_i$ and the test point $(X_{n+1}, Y_{n+1}(1) - Y_{n+1}(0))$. Predictive intervals that satisfy the coverage condition in (3) are said to be *marginally valid*.

## 2.2 Conformal prediction

Conformal prediction (CP) is a model- and distribution-free framework for predictive inference that provides (finite-sample) marginal coverage guarantees. In what follows, we describe a variant of CP, known as *split* (or *inductive*) CP [12, 13, 14], for the standard regression setup. Given a training dataset $\mathcal{D} = \{(X_i, Y_i)\}_i$, the CP procedure starts by splitting $\mathcal{D}$ into two disjoint subsets: a proper training set $\{(X_j, Y_j) : j \in \mathcal{D}_t\}$, and a *calibration* set $\{(X_k, Y_k) : k \in \mathcal{D}_c\}$. Then, an ML model $\widehat{\mu}(x)$ is fit using the samples in $\mathcal{D}_t$ and a *conformity score* $V(.)$ is evaluated for all samples in $\mathcal{D}_c$ as follows:

$$V_k(\widehat{\mu}) \triangleq V(X_k, Y_k; \widehat{\mu}), \ \forall k \in \mathcal{D}_c. \tag{4}$$

The conformity score measures how unusual the prediction looks relative to previous examples. A common choice of $V(.)$ is the absolute residual, i.e., $V(x, y; \widehat{\mu}) \triangleq |\widehat{\mu}(x) - y|$. For a target coverage level of $1 - \alpha$, we then compute a quantile of the empirical distribution of conformity scores, i.e.,

$$Q_{\mathcal{V}}(1 - \alpha) \triangleq (1 - \alpha)(1 + 1/|\mathcal{D}_c|)\text{-th quantile of } \mathcal{V}(\widehat{\mu}), \tag{5}$$

where $\mathcal{V}(\widehat{\mu}) = \{V_k(\widehat{\mu}) : k \in \mathcal{D}_c\}$. Finally, the predictive interval at a new point $X_{n+1} = x$ is

$$\widehat{C}(x) = [\widehat{\mu}(x) - Q_{\mathcal{V}}(1 - \alpha), \widehat{\mu}(x) + Q_{\mathcal{V}}(1 - \alpha)]. \tag{6}$$

The interval in (6) is guaranteed to satisfy marginal coverage, i.e., $\mathbb{P}(Y_{n+1} \in \widehat{C}(X_{n+1})) \geq 1 - \alpha$. The only assumption needed for this condition to hold is the *exchangeability* between calibration and test data [12, 23, 24]. Note that the interval in (6) has a fixed length of $2Q_{\mathcal{V}}(1 - \alpha)$ that is independent of $x$. To enable adaptive intervals, [25] proposed a variant of the CP procedure where the base model is a quantile regression with interval-valued predictions $[\widehat{\mu}_{\alpha/2}(x), \widehat{\mu}_{1-\alpha/2}(x)]$, and the conformity score is defined as the signed distance $V_k(\widehat{\mu}) \triangleq \max\{\widehat{\mu}_{\alpha/2}(X_k) - Y_k, Y_k - \widehat{\mu}_{1-\alpha/2}(X_k)\}$.

## 2.3 Oracle conformal prediction of ITEs

How can we adapt the CP framework for predictive inference of ITEs? In a hypothetical world where we have access to counterfactual outcomes, we can apply the standard CP in Section 2.2 to a dataset of covariates and ITE tuples, $\mathcal{D}^* = \{(X_i, Y_i(1) - Y_i(0))\}_i$, and compute conformity scores as:

$$V_k^*(\widehat{\tau}) \triangleq V(X_k, Y_k(1) - Y_k(0); \widehat{\tau}), \forall k \in \mathcal{D}_c^*, \tag{7}$$

where $\widehat{\tau}$ is an ML model fit to estimate the CATE function $\tau(x)$ using $\mathcal{D}_t^*$, and $\mathcal{D}^* = \mathcal{D}_t^* \cup \mathcal{D}_c^*$. We will refer to this procedure as "oracle" conformal prediction and to $V_k^*(\widehat{\tau})$ as the oracle conformity scores. Since the oracle problem is a standard regression, the oracle procedure is marginally valid—i.e., it satisfies the guarantee in (3), $\mathbb{P}(Y(1) - Y(0) \in \widehat{C}^*(X)) \geq 1 - \alpha$. However, oracle CP is infeasible since we can only observe one of the potential outcomes (colored in red and blue in (7)), hence we need an alternative procedure that operates only on the observed variable $Z = (X, W, Y)$.

## 2.4 The two challenges of predictive inference on ITEs

A naïve approach to inference of ITEs is to split the observed sample $\{Z_i = (X_i, W_i, Y_i)\}_i$ by the treatment group and create two datasets: $\mathcal{D}_0 = \{(X_i, Y_i) : W_i = 0\}_i$, $\mathcal{D}_1 = \{(X_i, Y_i) : W_i = 1\}_i$, then generate two sets of conformity scores for the nuisance estimates $\widehat{\mu}_0$ and $\widehat{\mu}_1$ as follows:

$$V_k^{(0)}(\widehat{\mu}_0) \triangleq V(X_k, Y_k(0); \widehat{\mu}_0), \forall k \in \mathcal{D}_{c,0}, \qquad V_k^{(1)}(\widehat{\mu}_1) \triangleq V(X_k, Y_k(1); \widehat{\mu}_1), \forall k \in \mathcal{D}_{c,1}, \tag{8}$$

where $\mathcal{D}_{c,0}$ and $\mathcal{D}_{c,1}$ are calibration subsets of $\mathcal{D}_0$ and $\mathcal{D}_1$. In order to construct valid predictive intervals for ITE using the conformity scores in (8), we need to reconsider how the two distinct characteristics of CATE estimation, previously discussed in Section 1, interact with the CP procedure:

### *(1) Covariate shift*

The distributions of covariates for treated and untreated subjects differ from that of the target population: $P_{X|W=0} \neq P_{X|W=1} \neq P_X$, i.e., the following holds for the conformity scores in (8):

$$P_{X,V^{(0)}|W=0} \neq P_{X,V^{(0)}} \qquad P_{X,V^{(1)}|W=1} \neq P_{X,V^{(1)}}$$

### *(2) Inductive biases*

Different choices for the joint model of $\mu_0$ and $\mu_1$ encode different inductive biases that impose different forms of regularization on the implied CATE function, i.e., $\widehat{\tau}(x) = \widehat{\mu}_1(x) - \widehat{\mu}_0(x)$ [5]. These biases influence the induced distributions $P_{V^{(0)}}$ and $P_{V^{(1)}}$ of the conformity scores in (8).

Covariate shift breaks the exchangeability assumption necessary for the validity of CP. Current methods have primarily focused on **(1)** with $Y(0)$ and $Y(1)$ as inference targets, and developed approaches for handling covariate shift by reweighting conformity scores [1, 26]. The resulting intervals for POs are then combined to produce intervals for ITEs. However, these method tie the CP procedure to model architecture, requiring inference on nuisance parameters, and hence lose the desirable post-hoc nature of CP. Furthermore, inference on POs is likely to provide conservative ITE intervals, and limits the inductive priors that can be assumed since not all CATE models provide explicit PO estimates.

# 3 Conformal Meta-learners

In [5], a taxonomy of "meta-learners" was introduced to categorize different inductive priors that can be incorporated into CATE estimators by structuring the regression models for $\mu_0$ and $\mu_1$. For example, the *T-learner* estimates $\widehat{\mu}_0$ and $\widehat{\mu}_1$ independently using $\mathcal{D}_0$ and $\mathcal{D}_1$, while the *S-learner* models the treatment variable $W$ as an additional covariate in a joint regression model $\widehat{\mu}(X, W)$ and estimates CATE as $\widehat{\tau}(x) = \widehat{\mu}(x, 1) - \widehat{\mu}(x, 0)$. In this Section, we propose an end-to-end solution to **(1)** and **(2)** by applying CP on top of CATE meta-learners in a post-hoc fashion, thereby decoupling the CP procedure from the CATE model and allowing direct inference on ITEs. In the next Section, we develop a unified framework for analyzing the validity of this broad class of procedures.

## 3.1 Pseudo-outcome regression for CATE estimation

We focus on a broad subclass of CATE meta-learners based on two-stage *pseudo-outcome* regression. These models replace the (unobserved) oracle ITEs with "proximal" targets that are estimated from observed variables only, and then train an ML model to predict the estimated targets from covariates. The two stages of this general pseudo-outcome regression framework can be described as follows:

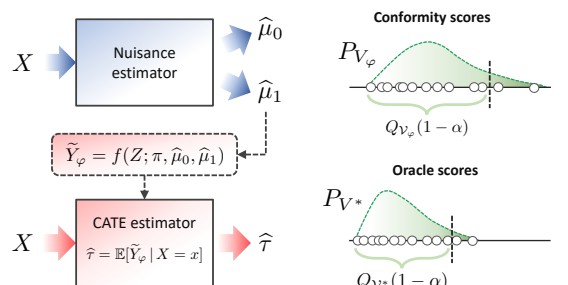

(a) Pseudo-outcome regression for CATE

(b) Conformal pseudo-intervals for ITE

**Conformity scores**

$P_{V_\varphi}$

$Q_{\mathcal{V}_\varphi}(1-\alpha)$

**Oracle scores**

$P_{V^*}$

$Q_{\mathcal{V}^*}(1-\alpha)$

| **Algorithm 1: Conformal Meta-learner** |
| --- |
| **Input** : $\mathcal{D} = \{(X_i, W_i, Y_i)\}_{i=1}^n$, $X_{n+1}$ 
 **Output** : $\widehat{C}_\varphi(X_{n+1})$ |
| **1** *Split $\mathcal{D}$ into $\mathcal{D}_\varphi$, $\mathcal{D}_t$ and $\mathcal{D}_c$;* |
| **2** **Pseudo-outcome regression:** |
| **3** **(1)** Estimate $\widehat{\varphi} = (\pi, \widehat{\mu}_0, \widehat{\mu}_1)$ using $\mathcal{D}_\varphi$; |
| **4** **(2)** Regress $\widetilde{Y}_\varphi = f(Z, \widehat{\varphi})$ on $X$ using $\mathcal{D}_t$; |
| **5** **Conformal pseudo-intervals:** |
| **6** Evaluate conformity scores $\mathcal{V}_\varphi$ using $\mathcal{D}_c$; |
| **7** **Return** $\widehat{C}_\varphi = [\widehat{\tau}(X_{n+1}) \pm Q_{\mathcal{V}_\varphi}(1-\alpha)]$ |

Figure 1: Pictorial depiction of conformal meta-learners.

**Stage 1.** We obtain a plug-in estimate $\widehat{\varphi}$ of the nuisance parameters $\varphi = (\pi, \mu_0, \mu_1)$. Note that since we assume that the propensity score is known, we only need to estimate $\mu_0$ and $\mu_1$ using $\mathcal{D}_0$ and $\mathcal{D}_1$.

**Stage 2.** We use the nuisance estimates obtained in Stage 1 to create pseudo-outcomes $\widetilde{Y}_\varphi$ that depend only on $\widehat{\varphi}$ and the observable variables $Z = (X, W, Y)$, i.e., $\widetilde{Y}_\varphi = f(Z, \widehat{\varphi})$ for some function $f$. The CATE estimate is then obtained by regressing the pseudo-outcome $\widetilde{Y}_\varphi$ on the covariate $X$. This is typically conducted using a different dataset than the one used to obtain the nuisance estimate $\widehat{\varphi}$.

The general framework described above captures various models in previous literature. We study 3 examples of such meta-learners: X-learner, Inverse propensity weighted (IPW) learner and doubly-robust (DR) learner.

| | **Pseudo-outcome** |
| --- | --- |
| *IPW-learner* [27] | $\widetilde{Y}_\varphi = \frac{W - \pi(X)}{\pi(X)(1-\pi(X))} Y$ |
| *X-learner* [5] | $\widetilde{Y}_\varphi = W(Y - \widehat{\mu}_0(X)) + (1 - W)(\widehat{\mu}_1(X) - Y)$ |
| *DR-learner* [20] | $\widetilde{Y}_\varphi = \frac{W - \pi(X)}{\pi(X)(1-\pi(X))}(Y - \widehat{\mu}_W(X)) + \widehat{\mu}_1(X) - \widehat{\mu}_0(X)$ |

Table 1: Existing meta-learners as instantiations of pseudo-outcome regression.

Table 1 lists the pseudo-outcomes $\widetilde{Y}_\varphi$ for the three meta-learners: IPW-learner reweights factual outcomes using propensity scores to match CATE, i.e., $\mathbb{E}[\widetilde{Y}_\varphi \mid X = x] = \tau(x)$; X-learner uses regression adjustment to impute counterfactuals; DR-learner combines both approaches. DR- and X-learners[1], coupled with specific architectures for joint modeling of $\widehat{\mu}_0$ and $\widehat{\mu}_1$, have shown competitive performance for CATE estimation in previous studies [5, 16, 20]. The conformal meta-learner framework decouples the CP procedure (Section 3.2) from the inductive priors encoded by these meta-learners, hence it inherits their favorable CATE estimation properties and enables a potentially more efficient direct inference on ITEs as opposed to inference on POs. This addresses **challenge (2)** in Section 2.4.

### 3.2 Conformal pseudo-intervals for ITEs

Pseudo-outcome regression is based on the notion that accurate proxies for treatment effects can produce reliable CATE point estimates. This concept can be extended to predictive inference: using CP to calibrate meta-learners via held-out pseudo-outcomes can yield accurate "pseudo-intervals" for ITEs.

Given a dataset $\mathcal{D} = \{Z_i = (X_i, W_i, Y_i)\}_i$, we create three mutually-exclusive subsets: $\mathcal{D}_\varphi$, $\mathcal{D}_t$ and $\mathcal{D}_c$. $\mathcal{D}_\varphi$ is used to estimate the nuisance parameters $\varphi$. Next, the estimates $\widehat{\varphi} = (\pi, \widehat{\mu}_0, \widehat{\mu}_1)$ are used to transform $\{Z_i = (X_i, W_i, Y_i) : i \in \mathcal{D}_t\}$ into covariate/pseudo-outcome pairs $\{(X_i, \widetilde{Y}_{\varphi,i}) : i \in \mathcal{D}_t\}$ which are used to train a CATE model $\widehat{\tau}$. Finally, we compute conformity scores for $\widehat{\tau}$ on pseudo-outcomes, i.e.,

$$V_{\varphi,k}(\widehat{\tau}) \triangleq V(X_k, \widetilde{Y}_{\varphi,k}; \widehat{\tau}), \ \forall k \in \mathcal{D}_c. \tag{9}$$

For a target coverage of $1 - \alpha$, we construct a predictive interval at a new point $X_{n+1} = x$ as follows:

$$\widehat{C}_\varphi(x) = [\widehat{\tau}(x) - Q_{\mathcal{V}_\varphi}(1-\alpha), \widehat{\tau}(x) + Q_{\mathcal{V}_\varphi}(1-\alpha)], \tag{10}$$

where $\mathcal{V}_\varphi = \{V_{\varphi,k}(\widehat{\tau}) : k \in \mathcal{D}_c\}$. We call $\widehat{C}_\varphi(x)$ a *pseudo-interval*. The conformal meta-learner approach is depicted in Figure 1 and a summary of the procedure is given in Algorithm 1.

---

[1]Here, we consider a special case of the X-learner in [5] which involves a weighted sum of two regression adjusted models $\widehat{\tau}_0$ and $\widehat{\tau}_1$ trained separately on the treated and control datasets $\mathcal{D}_0$ and $\mathcal{D}_1$.

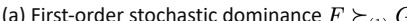
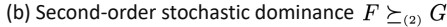
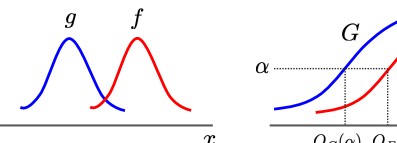
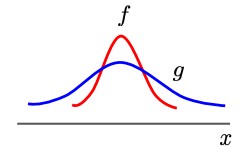
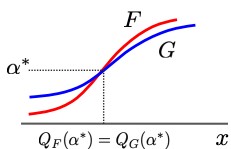

Figure 2: Graphical illustration of stochastic dominance among two exemplary distributions $F$ and $G$.

Note that conditional on $\widehat{\varphi}$, the pseudo-outcomes $(X, \widetilde{Y}_\varphi)$ in calibration data are drawn from the target distribution, which maintains the exchangeability of conformity scores and addresses covariate shift (**challenge (1)** in Section 2.4). However, the conformity scores $V\varphi$ are evaluated on transformed outcomes, which means that $V_\varphi$ and $V^*$ are not exchangeable, even though they are drawn from the same covariate distribution. Consequently, the usual CP guarantees, i.e., $\mathbb{P}(\widetilde{Y}_\varphi \in \widehat{C}_\varphi(X)) \geq 1 - \alpha$, do not immediately translate to coverage guarantees for the true ITE $Y(1) - Y(0)$. In the next section, we show that for certain choices of the pseudo-outcomes, the corresponding pseudo-intervals can provide valid inferences for ITE without requiring the exchangeability of $V_\varphi$ and $V^*$.

## 4 Validity of Conformal Meta-learners: A Stochastic Ordering Framework

Under what conditions are pseudo-intervals valid for inference of ITEs? Recall that these intervals are constructed by evaluating the empirical quantile of pseudo-outcome conformity scores. Intuitively, the pseudo-intervals will cover the true ITE if the conformity scores are "stochastically larger" than the oracle scores in Section 2.3, i.e., $Q_{\mathcal{V}_\varphi}(\alpha) \geq Q_{\mathcal{V}^*}(\alpha)$ in some stochastic sense (Figure 1(b)). Hence, to study the validity of conformal meta-learners, we analyze the *stochastic orders* of $V_\varphi$ and $V^*$, and identify conditions under which pseudo-intervals cover oracle intervals.

Stochastic orders are partial orders of random variables used to compare their location, magnitude, or variability [28, 29]. In our analysis, we utilize two key notions of stochastic order among cumulative distribution functions (CDFs) $F$ and $G$, which we formally define below.

**Definition 1 (Stochastic dominance)** *$F$ has first-order stochastic dominance (FOSD) on $G$, $F \succeq_{(1)} G$, iff $F(x) \leq G(x), \forall x$, with strict inequality for some $x$. $F$ has second-order stochastic dominance (SOSD) over $G$, $F \succeq_{(2)} G$, iff $\int_{-\infty}^{x} [G(t) - F(t)]\, dt \geq 0, \forall x$, with strict inequality for some $x$.*

**Definition 2 (Convex dominance)** *$F$ has monotone convex dominance (MCX) over $G$, $F \succeq_{mcx} G$, iff $\mathbb{E}_{X \sim F}[u(X)] \geq \mathbb{E}_{X \sim G}[u(X)]$ for all non-decreasing convex functions $u : \mathbb{R} \to \mathbb{R}$.*

Stochastic ordering is useful tool in decision theory and quantitative finance used to analyze the decisions of utility maximizers with varying risk attitudes [30]. A distribution $F$ has FOSD over $G$ if it is favored by any decision-maker with a non-decreasing utility function, i.e., $F$ is more likely to give higher outcomes than $G$ because its CDF is strictly lower (Figure 2(a)). If $F$ has SOSD over $G$, then it is favored by risk-averse decision-makers, i.e., $f$ has smaller spread than $g$ and is favored by all decision-makers with a non-decreasing *concave* utility function [31]. In this case, the CDFs can cross but $G$ is always lower after the last crossing point (Figure 2(b)). $F$ has MCX over $G$ if it is favored by decision-makers with a non-decreasing *convex* utility—in this case, the CDFs can cross but $F$ is always lower after the last crossing point (See Appendix A for a detailed analysis).

In the following Theorem, we provide sufficient conditions for the validity of conformal meta-learners in terms of the stochastic orders of their conformity scores.

**Theorem 1.** *If $(X_i, W_i, Y_i(0), Y_i(1))$, $i = 1, \ldots, n + 1$ are exchangeable, then $\exists \alpha^* \in (0, 1)$ such that the pseudo-interval $\widehat{C}_\varphi(X_{n+1})$ constructed using the dataset $\mathcal{D} = \{(X_i, W_i, Y_i)\}_{i=1}^{n}$ satisfies*

$$\mathbb{P}(Y_{n+1}(1) - Y_{n+1}(0) \in \widehat{C}_\varphi(X_{n+1})) \geq 1 - \alpha, \forall \alpha \in (0, \alpha^*),$$

*if at least one of the following stochastic ordering conditions hold: (i) $V_\varphi \succeq_{(1)} V^*$, (ii) $V_\varphi \preceq_{(2)} V^*$, and (iii) $V_\varphi \succeq_{mcx} V^*$. Under condition (i), we have $\alpha^* = 1$.*

All proofs are provided in Appendix A. Theorem 1 states that if the conformity score $V_\varphi$ of the meta-learner is stochastically larger (FOSD) or has a larger spread (SOSD and MCX) than the oracle conformity score, then the conformal meta-learner is valid for high-probability coverage (Figure 3). (This is the range of target coverage that is of practical relevance, i.e., $\alpha$ is typically set to 0.05 or 0.1.)

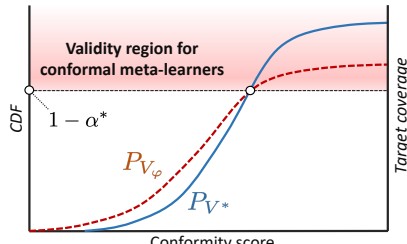

Figure 3: Validity condition in Theorem 1.

| Meta-learner | Conformity score | |
| --- | --- | --- |
| | *Absolute residual* | *Signed distance* |
| *X-learner* | No stochastic order | No stochastic order |
| *IPW-learner* | $V_\varphi \succeq_{mcx} V^*$ | $V_\varphi \preceq_{(2)} V^*$ |
| *DR-learner* | $V_\varphi \succeq_{mcx} V^*$ | $V_\varphi \preceq_{(2)} V^*$ |

Table 2: Stochastic orders of conformity scores for the three meta-learners considered in Table 1.

Because stochastic (or convex) dominance pertain to more variable conformity scores, the predictive intervals of conformal meta-learners will naturally be more conservative than the oracle intervals.

Whether a meta-learner meets conditions *(i)-(iii)* of Theorem 1 depends on how the pseudo-outcome, $\widetilde{Y}_\varphi = f(Z; \widehat{\varphi})$, is constructed. The following Theorem provides an answer to the question of which of the meta-learners listed in Table 1 satisfy the stochastic ordering conditions in Theorem 1.

**Theorem 2.** *Let $V_\varphi(\widehat{\tau}) = |\widehat{\tau}(X) - \widetilde{Y}_\varphi|$ and assume that the propensity score function $\pi : \mathcal{X} \to [0, 1]$ is known. Then, the following holds: (i) For the X-learner, $V_\varphi$ and $V^*$ do not admit to a model- and distribution-free stochastic order, (ii) For any distribution $P(X, W, Y(0), Y(1))$, CATE estimate $\widehat{\tau}$, and nuisance estimate $\widehat{\varphi}$, the IPW- and the DR-learners satisfy $V_\varphi \succeq_{mcx} V^*$.*

Theorem 2 states that the stochastic ordering of $V_\varphi$ and $V^*$ depends on the specific choice of the conformity score function $V(X, \widetilde{Y}_\varphi; \widehat{\tau})$ as well as the choice of the meta-learner, i.e., the pseudo-outcome generation function $\widetilde{Y}_\varphi = f(Z; \widehat{\varphi})$. The IPW- and DR-learners ensure that, by construction, the pseudo-outcome is equal to CATE in expectation: $\mathbb{E}[\widetilde{Y}_\varphi \mid X = x] = \tau(x)$. This construction enables the IPW- and DR-learners to provide unbiased estimates of average treatment effects (ATE) independent of the data distribution and the ML model used for the nuisance estimates $\widehat{\mu}_0$ and $\widehat{\mu}_1$. By the same logic, IPW- and DR-learners also guarantee stochastic (convex) dominance of their conformity scores irrespective of the data distribution and the ML model choice, hence preserving the model- and distribution-free nature of the CP coverage guarantees. Contrarily, the X-learner does not use the knowledge of $\pi$ to construct its pseudo-outcomes, hence it does not guarantee a (distribution-free) stochastic order and the achieved coverage depends on the nuisance estimates $\widehat{\mu}_0$ and $\widehat{\mu}_1$. In Table 2, we list the stochastic orders achieved for different choices of meta-learners and conformity scores. (The analysis of stochastic orders for the signed distance score used in [25] and [32] is provided in Appendix A.)

**Key limitations of conformal meta-learners.** While conformalized meta-learners can enable valid end-to-end predictive inference of ITEs, they have two key limitations. First, the propensity score $\pi$ must be known to guarantee model- and distribution-free stochastic ordering of conformity scores. However, we note that this limitation in not unique to our method and is also encountered in methods based on weighted CP [1, 26]. The second limitation is peculiar to our method: exact characterization of $\alpha^*$ is difficult and depends on the data distribution. Devising procedures for inferring $\alpha^*$ based on observable variables or deriving theoretical upper bounds on $\alpha^*$ are interesting directions for future work. Here, we focus on empirical evaluation of $\alpha^*$ in semi-synthetic experiments. A detailed comparison between our method and previous work is provided in Appendix B.

## 5 Experiments

### 5.1 Experimental setup

Since the true ITEs are never observed in real-world datasets, we follow the common practice of conducting numerical experiments using synthetic and semi-synthetic datasets [1, 8, 19]. We present a number of representative experiments in this Section and defer further results to Appendix C.

**Synthetic datasets.** We consider a variant of the data-generation process in Section 3.6 in [1] which was originally proposed in [7]. We create synthetic datasets by sampling covariates $X \sim U([0, 1]^d)$ and treatments $W|X = x \sim \text{Bern}(\pi(x))$ with $\pi(x) = (1 + I_x(2, 4))/4$, where $I_x(2, 4)$ is the regularized incomplete beta function (i.e., CDF of a Beta distribution with shape parameters 2 and 4). Outcomes are modeled as $\mu_1(x) = \zeta(x_1) \cdot \zeta(x_2)$ and $\mu_0(x) = \gamma \zeta(x_1) \cdot \zeta(x_2)$, where $\gamma \in [0, 1]$ is a parameter that controls the treatment effect, and $\zeta$ is a function given by $\zeta(x) = 1/(1 + \exp(-12(x - 0.5)))$.

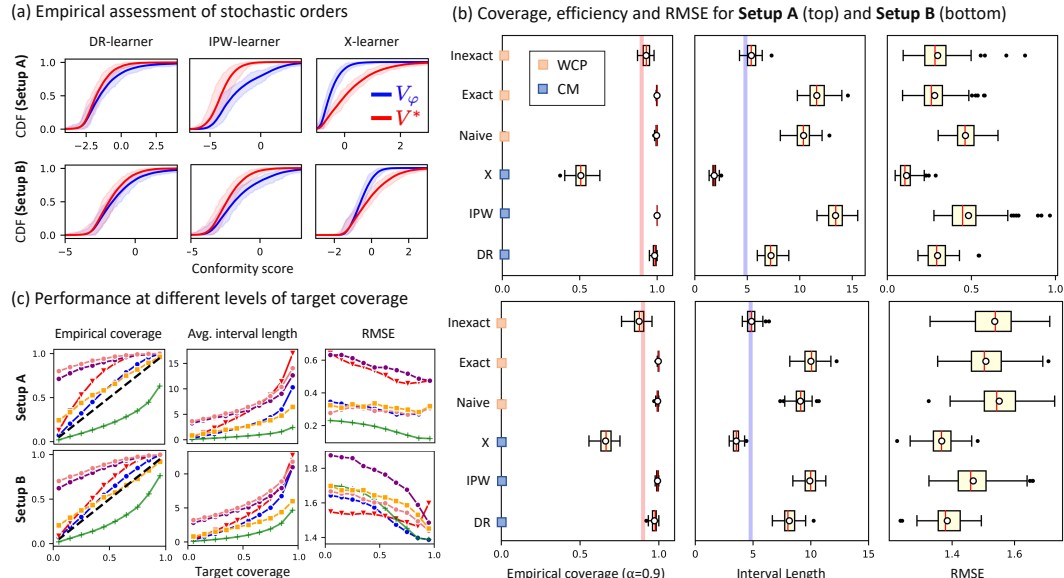

Figure 4: Performance of all baseline in the synthetic setup described in Section 5.1. In (b), red vertical lines correspond to target coverage $(1 - \alpha = 0.9)$, and blue vertical lines correspond to optimal interval width. In (c), baseline methods are color-coded as follows: ● CM-DR, ● CM-IPW, ● CM-X, ● WCP-Naïve, ● WCP-Exact, and ● WCP-Inexact. Here, WCP stands for weighted CP and CM stands for conformal meta-learners.

We assume that POs are sampled from $Y(w)|X = x \sim \mathcal{N}(\mu_w(x), \sigma^2(x))$, $w \in \{0, 1\}$ and consider a heteroscedastic noise model $\sigma^2(x) = -\log(x_1)$. We define two setups within this model: **Setup A** where the treatment has not effect $(\zeta = 1)$, and **Setup B** where the effects are heterogeneous $(\zeta = 0)$.

**Semi-synthetic datasets.** We also consider two well-known semi-synthetic datasets that involve real covariates and simulated outcomes. The first is the National Study of Learning Mindsets (NLSM) [3], and the second is the IHDP benchmark originally developed in [8]. Details on the data generation process for NLSM can be founded in Section 2 in [33]. Details on the IHDP benchmark can be found in [6, 8, 16, 19]. Appendix C provides detailed description of both datasets for completeness.

**Baselines.** We consider baseline models that provide valid predictive intervals for ITEs. Specifically, we consider state-of-the-art methods based on weighted conformal prediction (WCP) proposed in [1]. These methods apply weighted CP to construct intervals for the two POs or plug-in estimates of ITEs. We consider the three variants of WCP in [1]: (1) *Naïve WCP* which combines the PO intervals using Bonferroni correction, (2) *Exact Nested WCP* which applies WCP to plug-in estimates of ITEs in treatment and control groups followed by a secondary CP procedure, and (3) *Inexact Nested WCP* which follows the same steps of the exact version but replaces the secondary CP with conditional quantile regression. (Note that Inexact Nested WCP does not provide coverage guarantees.) For all baselines, we use the same model (Gradient Boosting) for nuisance and pseudo-outcome modeling, and we use the conformal quantile regression method in [25] to construct predictive intervals.

## 5.2 Results and discussion

Our experimental findings yield the following key takeaways: Firstly, the *IPW- and DR-learners demonstrate a robust FOSD (i.e., $\alpha^* = 1$) in the majority of experiments, surpassing the MCX conditions outlined in Theorem 2.* Secondly, *the DR-learner exhibits superior point estimation accuracy and interval efficiency in most experiment* compared to all other baselines that ensure valid inference. Thirdly, the *effectiveness of conformal meta-learners depends on the discrepancy between the CDFs of conformity scores and oracle scores*—pseudo-outcome transformations that induce thicker tails in the resulting conformtiy scores can cause conformal meta-learners to under-perform.

**Empirical assessment of stochastic orders.** Figure 4(a) depicts the empirical CDF of the conformity scores $V_\varphi$ and oracle scores $V^*$ for the three meta-learners under study (DR-, IPW- and X-learners). These CDFs are averaged over 100 runs of **Setups A** and **B** of the synthetic generation process outlined in Section 5.1. (The shaded regions represent the lowest and highest bounds on the empirical

|  | IHDP | | | NLSM | | |
|---|---|---|---|---|---|---|
|  | Coverage | Avg. len. | RMSE | Coverage | Avg. len. | RMSE |
| Naïve | 0.89 (0.02) | 18.9 (4.04) | 4.73 (1.00) | 0.99 (0.00) | 4.82 (0.02) | 0.15 (0.00) |
| Exact | 0.99 (0.00) | 29.8 (7.60) | 4.50 (0.97) | 0.99 (0.00) | 4.92 (0.07) | 0.19 (0.01) |
| Inexact | 0.61 (0.04) | 8.49 (1.36) | 4.61 (0.99) | 0.96 (0.00) | **2.38** (0.01) | 0.18 (0.00) |
| X | 0.65 (0.04) | 11.0 (3.04) | **3.34** (0.56) | 0.27 (0.00) | 0.38 (0.00) | **0.14** (0.00) |
| IPW | 0.99 (0.00) | 112 (23.0) | 19.9 (3.44) | 0.99 (0.00) | 6.48 (0.07) | 0.61 (0.02) |
| DR | 0.96 (0.01) | **16.7** (3.30) | **3.32** (0.53) | 0.99 (0.00) | 6.24 (0.07) | 0.37 (0.01) |

Table 3: Performance of all baselines in semi-synthetic datasets. Bold numbers correspond to best performance.

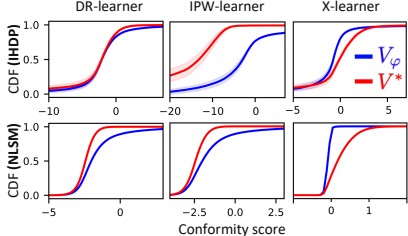

Figure 5: Stochastic orders in IHDP/NLSM.

CDFs evaluated across all runs.) In both setups, the conformity scores for the DR- and IPW-learners demonstrate FOSD over the oracle scores with respect to the average CDFs and in almost all realizations. This aligns with the result of Theorem 2, and shows that the stochastic dominance condition achieved in practice is even stronger than our theoretical guarantee since FOSD ($V_\varphi \succeq_{(1)} V^*$) implies the weaker conditions of $V_\varphi \preceq_{(2)} V^*$ and $V_\varphi \succeq_{mcx} V^*$. On the contrary, the conformity scores of the X-learner are dominated by oracle scores in the FOSD sense. This is not surprising in light of Theorem 2, which indicates that X-learners do not guarantee a distribution-free stochastic order. These observations are also replicated in the semi-synthetic datasets as shown in Figure 5.

Based on Theorem 1, the empirical stochastic orders observed in Figures 4(a) and 5 predict that the IPW- and DR-learners will cover ITEs, whereas the X-learner will not achieve coverage. This is confirmed by the results in Figures 4(b), 4(c) and Table 3. The fact that the IPW- and DR-learners satisfy a stronger FOSD condition is promising because it indicates that the range of validity for these models spans all levels of coverage ($\alpha^* = 1$ in Figure 3). It also means that a stronger version of Theorem 2 outlining the conditions under which IPW- and DR-learners achieve FOSD could be possible.

**Coverage, efficiency and point estimation accuracy.** The performance of a predictive inference procedure can be characterized in terms of three metrics: achieved coverage for true ITEs, expected length of predictive intervals, and root-mean-square error (RMSE) in CATE estimates. In most experiments, we find that the DR-learner strikes a balance between these metrics (See Appendix C for further experiments). In Figure 4(b), we can see that the DR-learner outperforms the valid (naïve and exact) WCP procedures in terms of RMSE and interval length, while achieving the target coverage of 90%. The X-learner outperforms all baselines in terms of RMSE, but as expected, it under-covers ITEs in all experiments. The inexact WCP baseline offers competitive efficiency and calibration, however, in addition to not offering coverage guarantees it also lacks consistency in RMSE performance under different inductive biases (i.e., no treatment effects in **Setup A** and heterogeneous effects in **Setup B**). These performance trends hold true across all levels of target coverage as shown in Figure 4(c).

The semi-synthetic experiments on IHDP and NLSM datasets shed light on when meta-learners may perform poorly. The DR-learner outperforms all baselines on the IHDP dataset in terms of RMSE, interval efficiency, while achieving the desired coverage of 90%. However, we observe that empirical performance depends on how closely the CDF of conformity scores matches the oracle CDF. The DR-learner performance deteriorates when conformity scores have "very strong" dominance over oracle scores, as observed in the NLSM dataset (Figure 5, bottom). Conversely, when the CDF of conformity scores is a closer lower bound on the oracle CDF, the DR-learner performance is competitive (Figure 4 and Figure 5, top). This is intuitive because if the pseudo-outcome transformation induces significant variability in regression targets, it will result in a lower CDF, poorer accuracy of pseudo-outcome regression, and longer predictive intervals. This is why the DR-learner consistently outperforms the IPW-learner, as it provides a closer approximation of the oracle CDF. Future work could focus on analyzing the gap between pseudo-outcome and oracle score CDFs and designing pseudo-outcome transformations that optimize efficiency while preserving stochastic orders.

## 6 Conclusions

Estimation and inference of treatment effects is challenging because causal effects are not directly observable. In this paper, we developed a general framework for inference of treatment effects, dubbed conformal meta-learners, that is compatible with any machine learning model. Our framework inherits the model- and distribution-free validity of conformal prediction as well as the estimation accuracy of model-agnostic meta-learners of treatment effects. Additionally, we introduce a new theoretical framework based on stochastic ordering to assess the validity of our method, which can guide the development of new models optimized for both accurate estimation and valid inference.

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

# Appendix A: Technical Proofs

## A.1. Equivalent Definitions for Stochastic Orders

The following section introduces alternative definitions for stochastic dominance between two cumulative distribution functions (CDFs), denoted as $F$ and $G$. These definitions will be used in the proof for Theorem 2 (Section A.4.). Note that these definitions are equivalent to Definition 1 provided in the main text but are expressed in decision-theoretic terms.

**Definition A1 (First-order stochastic dominance)** $F$ has *first-order* stochastic dominance (FOSD) over $G$, $F \succeq_{(1)} G$, if and only if $\mathbb{E}_{X \sim F}[u(X)] \geq \mathbb{E}_{X \sim G}[u(X)]$ for all non-decreasing (utility) functions $u : \mathbb{R} \to \mathbb{R}$ (Theorem 1.A.3 in [28] proves the equivalence of Definitions 1 and A1).

**Definition A2 (Second-order stochastic dominance)** $F$ has *second-order* stochastic dominance (SOSD) over $G$, $F \succeq_{(2)} G$, if and only if $\mathbb{E}_{X \sim F}[u(X)] \geq \mathbb{E}_{X \sim G}[u(X)]$ for all non-decreasing concave functions $u : \mathbb{R} \to \mathbb{R}$ (Refer to Theorem 4.A.1 and Eq. (4.A.7) in [28]).

In addition to the alternative definitions of FOSD and SOSD above, we also state the definitions for two additional notions of stochastic ordering that we will use in the proofs.

**Definition A3 (Statewise stochastic dominance)** A random variable $X$ has *statewise* stochastic dominance (SWD) over a random variable $Y$, $X \succeq_{\text{SWD}} Y$, if and only if $X \geq Y$ for every draw of the bivariate random variable $(X, Y) \sim P_{X,Y}$. Note that SWD implies FOSD.

**Definition A4 (Convex dominance)** $F$ has *convex* dominance (CX) over $G$, $F \succeq_{\text{cx}} G$, if and only if $\mathbb{E}_{X \sim F}[u(X)] \geq \mathbb{E}_{X \sim G}[u(X)]$ for all convex functions $u : \mathbb{R} \to \mathbb{R}$.

## A.2. Useful Lemmas

**Lemma A1. (Pointwise and marginal stochastic orders)** *For random variables $X$, $Y$, and $Z$, the following conditions hold for any marginal distribution $P_X$:*

*(i)* $Y \mid X = x \ \succeq_{(1)} Z \mid X = x, \forall x \in \mathcal{X} \quad \Rightarrow \ Y \succeq_{(1)} Z,$

*(ii)* $Y \mid X = x \ \succeq_{(2)} Z \mid X = x, \forall x \in \mathcal{X} \quad \Rightarrow \ Y \succeq_{(2)} Z,$

*(iii)* $Y \mid X = x \ \succeq_{mcx} Z \mid X = x, \forall x \in \mathcal{X} \Rightarrow \ Y \succeq_{mcx} Z.$

*Proof.* Recall from the definition of FOSD (Definition A1) that if $Y \mid X = x \succeq_{(1)} Z \mid X = x$, then:

$$\mathbb{E}_{V \sim P_{Y|X=x}}[u(V)] \geq \mathbb{E}_{W \sim P_{Z|X=x}}[u(W)], \ \forall x \in \mathcal{X}, \tag{A.1}$$

for all non-decreasing functions $u : \mathbb{R} \to \mathbb{R}$. Since expectation is a positive linear operator, marginalizing both sides with respect to the density $P_X$ preserves the inequality in (A.1):

$$\mathbb{E}_{X \sim P_X}[\mathbb{E}_{V \sim P_{Y|X}}[u(V)]] \geq \mathbb{E}_{X \sim P_X}[\mathbb{E}_{W \sim P_{Z|X}}[u(W)]], \tag{A.2}$$

which recovers the definition of FOSD between random variables $Y$ and $Z$, i.e.,

$$\mathbb{E}_{Y \sim P_Y}[u(Y)] \geq \mathbb{E}_{Z \sim P_Z}[u(Z)], \tag{A.3}$$

and concludes the proof for (i). Note that (A.3) holds for all non-decreasing functions $u : \mathbb{R} \to \mathbb{R}$, which also includes all non-decreasing concave and convex functions that define SOSD and MCX. Hence, combining (A.1)-(A.3) with Definitions 2 and A2 concludes the proof for (ii) and (iii). $\square$

**Lemma A2. (Convex dominance of mixture distributions over convex combinations of random variables)** *Let $X \in \mathbb{R}$ and $Y \in \mathbb{R}$ be two real-valued random variables with finite means, and let $\pi \in [0, 1]$ be a constant. If $Z$ and $V$ are two random variables constructed as:*

- *$Z$ is the absolute value of the convex combination of $X$ and $Y$, i.e., $Z = |\pi X + (1 - \pi)Y|$,*
- *$V$ is a mixture of $|X|$ and $|Y|$, i.e., $V = W|X| + (1 - W)|Y|$, $W \sim Bernoulli(\pi)$,*

*then we have $V \succeq_{mcx} Z$ for any $\pi \in [0, 1]$ and any distribution $P(X, Y)$.*

*Proof.* Recall from Definition 2 that $V \succeq_{mcx} Z$ if and only if

$$\mathbb{E}_V[u(V)] \geq \mathbb{E}_Z[u(Z)], \tag{A.4}$$

for all non-decreasing convex functions $u : \mathbb{R} \to \mathbb{R}$. Given the definition of $V$ above, we have

$$
\begin{aligned}
\mathbb{E}_V[u(V)] &= P(W = 1) \cdot \mathbb{E}_V[u(V) \,|\, W = 1] + P(W = 0) \cdot \mathbb{E}_V[u(V) \,|\, W = 0] \\
&= \pi \cdot \mathbb{E}[u(|X|)] + (1 - \pi) \cdot \mathbb{E}[u(|Y|)] \\
&\overset{(a)}{\geq} \mathbb{E}[u(\pi|X| + (1 - \pi)|Y|)] = \mathbb{E}[u(|\pi X| + |(1 - \pi)Y|)] \\
&\overset{(b)}{\geq} \mathbb{E}[u(|\pi X + (1 - \pi)Y|)] = \mathbb{E}[u(Z)] \Rightarrow V \succeq_{mcx} Z,
\end{aligned}
\tag{A.5}
$$

where the last two inequalities follow from (a) the convexity of $u$, and (b) the monotonicity of $u$ along with the application of the triangle inequality $|\pi X| + |(1 - \pi)Y| \geq |\pi X + (1 - \pi)Y|$. $\qquad\square$

**Lemma A3. (Stochastic dominance and sign changes in CDFs)** *Let $X$ and $Y$ be two non-negative random variables with distribution functions $F$ and $G$ and with finite means such that $\mathbb{E}[X] \leq \mathbb{E}[Y]$. Let $S^-(f)$ be the number of sign changes of a function $f$ defined as*

$$
S^-(f) = \sup S^-[f(x_1), f(x_2), \ldots, f(x_m)],
\tag{A.6}
$$

*where $S^-[a_1, a_2, \ldots, a_m]$ is the number of sign changes of the sequence $[a_1, a_2, \ldots, a_m]$, with the zero terms being discarded, and the supremum in (A.6) is taken over all sets $x_1 < x_2 < \ldots < x_m$ such that $m < \infty$. Then, $F \succeq_{(2)} [\succeq_{mcx}]G$ if and only if there exist random variables $Z_1, Z_2, \ldots$, with distribution functions $G_1, G_2, \ldots$, such that $Z_1 =_{st} X$, $\mathbb{E}[Z_j] \leq \mathbb{E}[Y]$, $j = 1, 2, \ldots$, $Z_j \to_{st} Y$ and $\mathbb{E}[Z_j] \to \mathbb{E}[Y]$ as $j \to \infty$ and $S^-(G_{j+1} - G_j) = 1$ and the sign sequence is $\{+, -\}[\{-, +\}]$. Here, $=_{st}$ denotes equality in law and $\to_{st}$ denotes convergence in distribution.*

*Proof.* Refer to Theorems 4.A.22 and 4.A.23 in [28] for a full proof. $\qquad\square$

**Corollary A4. (CDF crossings under SOSD and MCX)** *Let $X$ and $Y$ be non-negative random variables with CDFs $F$ and $G$ with finite means such that $\mathbb{E}[X] \leq \mathbb{E}[Y]$. Then, if $F \succeq_{(2)} [\succeq_{mcx}]G$, we have $S^-(F - G) \geq 1$ and the corresponding sign sequence is $\{\ldots, -, +\}[\{\ldots, +, -\}]$. Equivalently, there exists $\alpha^* \in (0, 1)$ such that $F^{-1}(\alpha^*) = G^{-1}(\alpha^*) = v^*$ and $F(v) \geq [\leq]G(v), \forall v \geq v^*$.*

*Proof.* From Lemma A3, we know that if $F \succeq_{(2)} [\succeq_{mcx}]G$ then there exists a sequence of random variables $Z_1, Z_2, \ldots$, with distributions $G_1, G_2, \ldots$, such that $Z_1 =_{st} X$, $Z_j \to_{st} Y$, and the CDFs satisfy $S^-(G_{j+1} - G_j) = 1$ with a sign sequence $\{-, +\}[\{+, -\}]$.

Now observe that for any three integers $k < m < n$, if $S^-(G_m - G_k) = 1$ with a sign sequence of $\{-, +\}[\{+, -\}]$, and $S^-(G_n - G_m) = 1$ with a sign sequence $\{-, +\}[\{+, -\}]$, then by monotonicity of CDFs it follows that in the last point of crossing, $G_n$ crosses $G_k$ from below [above], i.e., $G_n - G_k$ has at least one sign change with $\{\ldots, -, +\}[\{\ldots, +, -\}]$. Thus, for any $j$, the three random variables $Z_1 =_{st} X$, $Z_j$ and $Y$ satisfy $S^-(G_j - F) = 1$ and $S^-(G - G_j) = 1$ with a sign sequence $\{\ldots, -, +\}[\{\ldots, +, -\}]$, which implies that in the last point of crossing, $F$ crosses $G$ from below [above] and concludes the statement of the corollary. $\qquad\square$

### A.3. Proof of Theorem 1

**Theorem 1.** *If $(X_i, W_i, Y_i(0), Y_i(1))$, $i = 1, \ldots, n + 1$ are exchangeable, then $\exists \alpha^* \in (0, 1)$ such that the pseudo-interval $\widehat{C}_\varphi(X_{n+1})$ constructed using the dataset $\mathcal{D} = \{(X_i, W_i, Y_i)\}_{i=1}^n$ satisfies*

$$
\mathbb{P}(Y_{n+1}(1) - Y_{n+1}(0) \in \widehat{C}_\varphi(X_{n+1})) \geq 1 - \alpha, \; \forall \alpha \in [0, \alpha^*],
$$

*if at least one of the following stochastic ordering conditions hold: (i) $V_\varphi \succeq_{(1)} V^*$, (ii) $V_\varphi \preceq_{(2)} V^*$, and (iii) $V_\varphi \succeq_{mcx} V^*$. Under condition (i), we have $\alpha^* = 1$.*

*Proof.* Without loss of generality, assume that the conformity scores are sorted, $V_{\varphi,1} < \ldots < V_{\varphi,n_c}$, where $n_c = |\mathcal{D}_c|$. Recall from (10) that the pseudo-interval is constructed as:

$$
\widehat{C}_\varphi(x) = [\widehat{\tau}(x) - Q_{\mathcal{V}_\varphi}(1 - \alpha), \widehat{\tau}(x) + Q_{\mathcal{V}_\varphi}(1 - \alpha)],
\tag{A.7}
$$

where $Q_{\mathcal{V}_\varphi}(1 - \alpha)$ is the empirical quantile defined as

$$
Q_{\mathcal{V}_\varphi}(1 - \alpha) = \begin{cases} V_{\varphi, \lceil (n_c+1)(1-\alpha) \rceil}, & \alpha \geq \frac{1}{n_c+1}, \\ \infty, & o.w. \end{cases}
\tag{A.8}
$$

Combining (A.7) and (A.8), we notice that the following two events are equivalent

$$\{Y_{n+1}(1) - Y_{n+1}(0) \in \widehat{C}_\varphi(X_{n+1})\} \iff \{V^*_{n+1} \leq Q_{\mathcal{V}_\varphi}(1-\alpha)\}, \tag{A.9}$$

or equivalently,

$$\{Y_{n+1}(1) - Y_{n+1}(0) \in \widehat{C}_\varphi(X_{n+1})\} \iff \{V^*_{n+1} \leq V_{\varphi, \lceil (n_c+1)(1-\alpha) \rceil}\}. \tag{A.10}$$

Hence, the achieved coverage probability for the pseudo-interval is given by

$$\mathbb{P}(V^*_{n+1} \leq V_{\varphi, \lceil (n_c+1)(1-\alpha) \rceil}). \tag{A.11}$$

By exchangeability of the variables $(X_1, Y_1(1) - Y_1(0)), ..., (X_{n+1}, Y_{n+1}(1) - Y_{n+1}(0))$, we have

$$\mathbb{P}(V_{\varphi, n+1} \leq V_{\varphi, k}) = \frac{k}{n_c + 1}, \tag{A.12}$$

for any integer $k$. (A.12) holds because ranks are uniformly distributed under exchangeability, i.e., $V_{\varphi, n+1}$ is equally likely to fall in anywhere between the calibration points $V_{\varphi, 1}, \ldots, V_{\varphi, n_c}$. From (A.12), it follows that for $\lceil (n_c + 1)(1-\alpha) \rceil$ we have:

$$\mathbb{P}(V_{\varphi, n+1} \leq V_{\varphi, \lceil (n_c+1)(1-\alpha) \rceil}) = \frac{\lceil (n_c + 1)(1-\alpha) \rceil}{n_c + 1} \geq 1 - \alpha. \tag{A.13}$$

Now, we examine the three conditions: (i) $V_\varphi \succeq_{(1)} V^*$, (ii) $V_\varphi \preceq_{(2)} V^*$, and (iii) $V_\varphi \succeq_{mcx} V^*$.

## (i) FOSD $V_\varphi \succeq_{(1)} V^*$:

If $V_\varphi \succeq_{(1)} V^*$, then from Definition 1: $F_{V_\varphi}(v) \leq F_{V^*}(v), \forall v$. Equivalently, FOSD can be written as:

$$V_\varphi \succeq_{(1)} V^* \iff \mathbb{P}(V_\varphi \leq v) \leq \mathbb{P}(V^* \leq v), \forall v. \tag{A.14}$$

Since (A.14) applies for any $v$, then the following holds:

$$V_{\varphi, n+1} \succeq_{(1)} V^*_{n+1} \iff \mathbb{P}(V_{\varphi, n+1} \leq V_{\varphi, \lceil (n_c+1)(1-\alpha) \rceil}) \leq \mathbb{P}(V^*_{n+1} \leq V_{\varphi, \lceil (n_c+1)(1-\alpha) \rceil}). \tag{A.15}$$

Combining (A.13) and (A.15) we have

$$V_{\varphi, n+1} \succeq_{(1)} V^*_{n+1} \Rightarrow \mathbb{P}(V^*_{n+1} \leq V_{\varphi, \lceil (n_c+1)(1-\alpha) \rceil}) \geq \mathbb{P}(V_{\varphi, n+1} \leq V_{\varphi, \lceil (n_c+1)(1-\alpha) \rceil}) \geq 1 - \alpha.,$$

which holds for any $\alpha$, hence $\alpha^* = 1$. This concludes statement (i).

## (ii) SOSD $V_\varphi \preceq_{(2)} V^*$:

If $V_\varphi \preceq_{(2)} V^*$, then from Corollary A4 we know that $\exists \alpha^* \in (0, 1)$ where $F^{-1}_{V_\varphi}(\alpha^*) = F^{-1}_{V^*}(\alpha^*) = v^*$ and $F_{V^*}(v) \geq F_{V_\varphi}(v), \forall v \geq v^*$. Equivalently, SOSD can be written as:

$$V_\varphi \preceq_{(2)} V^* \Rightarrow \exists v^*, \ s.t. \ \mathbb{P}(V_\varphi \leq v) \leq \mathbb{P}(V^* \leq v), \forall v \geq v^*. \tag{A.16}$$

Hence, the following holds:

$$V_{\varphi, n+1} \preceq_{(2)} V^*_{n+1} \Rightarrow \mathbb{P}(V_{\varphi, n+1} \leq v) \leq \mathbb{P}(V^*_{n+1} \leq v), \forall v \geq v^*,$$
$$\Rightarrow \mathbb{P}(V_{\varphi, n+1} \leq V_{\varphi, \lceil (n_c+1)(1-\alpha) \rceil}) \leq \mathbb{P}(V^*_{n+1} \leq V_{\varphi, \lceil (n_c+1)(1-\alpha) \rceil}), \tag{A.17}$$

for all $\alpha \leq \alpha^*$. Combining (A.13) and (A.17) we have

$$V_{\varphi, n+1} \preceq_{(2)} V^*_{n+1} \Rightarrow \mathbb{P}(V^*_{n+1} \leq V_{\varphi, \lceil (n_c+1)(1-\alpha) \rceil}) \geq \mathbb{P}(V_{\varphi, n+1} \leq V_{\varphi, \lceil (n_c+1)(1-\alpha) \rceil}) \geq 1 - \alpha.,$$

for all $0 \leq \alpha \leq \alpha^*$. This concludes statement (ii).

## (iii) MCX $V_\varphi \succeq_{mcx} V^*$:

Since Corollary A4 holds for SOSD and MCX, the proof is identical to the proof for (ii). □

### A.4. Proof of Theorem 2

**Theorem 2.** *Let $V_\varphi(\widehat{\tau}) = |\widehat{\tau}(X) - \widetilde{Y}_\varphi|$ and assume that the propensity score function $\pi : \mathcal{X} \to [0, 1]$ is known. Then, the following holds: (i) For the X-learner, $V_\varphi$ and $V^*$ do not admit to a model- and distribution-free stochastic order, (ii) For any distribution $P(X, W, Y(0), Y(1))$, CATE estimate $\widehat{\tau}$, and nuisance estimate $\widehat{\varphi}$, the IPW- and the DR-learners satisfy $V_\varphi \succeq_{mcx} V^*$.*

*Proof.* For a CATE estimate $\widehat{\tau}$, the oracle scores are

$$V^*(\widehat{\tau}) = |\widehat{\tau}(X) - (Y(1) - Y(0))|. \tag{A.18}$$

In what follows, we use the notation $V_\varphi(x)$ to denote the conditional random variable $V_\varphi|X = x$, i.e., the conformity score evaluated at a given feature point. Similarly, we use $V^*(x)$ to denote the conditional random variable $V^*|X = x$ (oracle score evaluated at a given feature point).

**Proof of statement (i):**

Recall from Table 1 that the pseudo-outcome for the X-learner is given by:

$$\widetilde{Y}_\varphi = W(Y - \widehat{\mu}_0(X)) + (1 - W)(\widehat{\mu}_1(X) - Y). \tag{A.19}$$

With $V_\varphi(\widehat{\tau}) = |\widehat{\tau}(X) - \widetilde{Y}_\varphi|$, the conformity score for the X-learner can be written as:

$$V_\varphi(\widehat{\tau}) = |\widehat{\tau}(X) - W(Y - \widehat{\mu}_0(X)) - (1 - W)(\widehat{\mu}_1(X) - Y)|. \tag{A.20}$$

To prove statement (i), it suffices to find counter-examples for estimators $\widehat{\tau}$, $\widehat{\mu}_0$ and $\widehat{\mu}_1$, or data distributions that cause violations of FOSD, SOSD and MCX between $V_\varphi$ and $V^*$. Let $u : \mathbb{R} \to \mathbb{R}$ be a non-decreasing function. The expected value of $u(V_\varphi(x))$ can be evaluated as:

$$\begin{aligned}
\mathbb{E}[u(V_\varphi(x))] &= \mathbb{E}[u(|\widehat{\tau}(x) - W(Y - \widehat{\mu}_0(x)) - (1 - W)(\widehat{\mu}_1(x) - Y)|)] \\
&= \pi(x) \cdot \mathbb{E}[u(|\widehat{\tau}(x) - (Y(1) - \widehat{\mu}_0(x))|)] + \\
&\quad (1 - \pi(x)) \cdot \mathbb{E}[u(|\widehat{\tau}(x) - (\widehat{\mu}_1(x) - Y(0))|)].
\end{aligned} \tag{A.21}$$

We start by showing that there exists nuisance and CATE estimates and a data distribution for which $V_\varphi$ does not have FOSD on $V^*$. Assume that $u$ is non-decreasing concave. By Jensen inequality, we have:

$$\mathbb{E}[u(V_\varphi(x))] \le \mathbb{E}[u(\pi(x) \cdot |\widehat{\tau}(x) - (Y(1) - \widehat{\mu}_0(x))| + (1 - \pi(x)) \cdot |\widehat{\tau}(x) - (\widehat{\mu}_1(x) - Y(0))|)].$$

Now consider a nuisance estimates of $\widehat{\mu}_0 = 0$ and $\widehat{\mu}_1 = 0$, a data distribution for which $Y(0) > 0$ and $Y(1) < 0$ almost surely, and a CATE estimator $\widehat{\tau}(x) > 0$, $\forall x \in \mathcal{X}$, then we have

$$\begin{aligned}
\mathbb{E}[u(V_\varphi(x))] &\le \mathbb{E}[u(\pi(x) \cdot |\widehat{\tau}(x) - Y(1)| + (1 - \pi(x)) \cdot |\widehat{\tau}(x) + Y(0)|)], \\
&= \mathbb{E}[u(\pi(x) \cdot (\widehat{\tau}(x) - Y(1)) + (1 - \pi(x)) \cdot (\widehat{\tau}(x) + Y(0)))], \\
&\le \mathbb{E}[u(\widehat{\tau}(x) - (Y(1) - Y(0)))], \\
&\le \mathbb{E}[u(|\widehat{\tau}(x) - (Y(1) - Y(0))|)] = \mathbb{E}[u(V^*(x))], 
\end{aligned} \tag{A.22}$$

and hence $V^*(x) \succeq_{(2)} V_\varphi(x)$ for all $x$, and by Lemma A1, we have $V^* \succeq_{(2)} V_\varphi$. The last inequality in (A.22) holds because $u$ is non-decreasing and $(1 - \pi(x))Y(1) - \pi(x)Y(0)$ is always non-positive when $Y(0) > 0$ and $Y(1) < 0$ (almost surely). Since SOSD is a necessary condition for FOSD, then it follows that $V_\varphi$ does not have FOSD over $V^*$ in this counter-example.

Now we show that there exists nuisance and CATE estimates and a data distribution for which $V_\varphi$ does not have MCX over $V^*$. This follows directly from the previous counter-example. Since $V^* \succeq_{(2)} V_\varphi$ implies that $-V_\varphi \succeq_{mcx} -V^*$ (Theorem 4.A.1. in [28]), then $V_\varphi$ does not have MCX over $V^*$.

Finally, we show that there exists nuisance and CATE estimates and a data distribution for which $V^*$ does not have SOSD over $V_\varphi$. Note that $\mathbb{E}[V^*] \ge \mathbb{E}[V_\phi]$ is a necessary condition for $V^* \succeq_{(2)} V_\varphi$ (See Eq. (4.A.6) in [28]). To compare $\mathbb{E}[V^*]$ and $\mathbb{E}[V_\phi]$, we rewrite $\mathbb{E}[V_\phi(x)]$ as:

$$\begin{aligned}
\mathbb{E}[V_\phi(x)] &= \mathbb{E}[|\widehat{\tau}(x) - W(Y - \widehat{\mu}_0(x)) - (1 - W)(\widehat{\mu}_1(x) - Y)|], \\
&= \pi(x) \cdot \mathbb{E}[|\widehat{\tau}(x) - (Y(1) - \widehat{\mu}_0(x))|] + (1 - \pi(x)) \cdot \mathbb{E}[|\widehat{\tau}(x) - (\widehat{\mu}_1(x) - Y(0))|], \\
&= \mathbb{E}[\pi(x) \cdot |\widehat{\tau}(x) - (Y(1) - \widehat{\mu}_0(x))| + (1 - \pi(x)) \cdot |\widehat{\tau}(x) - (\widehat{\mu}_1(x) - Y(0))|], \\
&\ge \mathbb{E}[|\widehat{\tau}(x) - \pi(x)(Y(1) - \widehat{\mu}_0(x)) - (1 - \pi(x))(\widehat{\mu}_1(x) - Y(0))|], \\
&= \mathbb{E}[|\widehat{\tau}(x) - (Y(1) - Y(0)) + (1 - \pi(x))(Y(1) - \widehat{\mu}_1(x)) + \pi(x)(\widehat{\mu}_0(x) - Y(0))|].
\end{aligned}$$

For nuisance estimates that satisfy $\widehat{\mu}_1(x) < Y(1)$ and $\widehat{\mu}_0(x) > Y(0)$ almost surely, it follows that $(1 - \pi(x))(Y(1) - \widehat{\mu}_1(x)) + \pi(x)(\widehat{\mu}_0(x) - Y(0))$ is always positive and hence we have:

$$
\begin{aligned}
\mathbb{E}[V_\phi(x)] &\geq \mathbb{E}[|\widehat{\tau}(x) - (Y(1) - Y(0)) + (1 - \pi(x))(Y(1) - \widehat{\mu}_1(x)) + \pi(x)(\widehat{\mu}_0(x) - Y(0))|], \\
&\geq \mathbb{E}[|\widehat{\tau}(x) - (Y(1) - Y(0))|] = \mathbb{E}[V^*(x)],
\end{aligned}
\tag{A.23}
$$

which implies that $V^*$ does not have SOSD over $V_\varphi$.

**Proof of statement (ii):**

From Table 1, the pseudo-outcomes for the IPW- and DR-learners can be written as:

$$
V_\varphi(x) = W \, |\widehat{\tau}(x) - Y_\pi(x)| + (1 - W) \, |\widehat{\tau}(x) - Y_{1-\pi}(x)|,
\tag{A.24}
$$

where $Y_\pi$ and $Y_{1-\pi}$ are defined as follows:

$$
Y_\pi(x) = \begin{cases} \frac{1}{\pi(x)} \, Y(1), & \text{for IPW-learner,} \\ \frac{1}{\pi(x)} \left(Y(1) - \widehat{\mu}_1(x)\right) + (\widehat{\mu}_1(x) - \widehat{\mu}_0(x)), & \text{for DR-learner, and} \end{cases}
$$

$$
Y_{1-\pi}(x) = \begin{cases} \frac{-1}{1-\pi(x)} \, Y(0), & \text{for IPW-learner,} \\ \frac{-1}{1-\pi(x)} \left(Y(0) - \widehat{\mu}_0(x)\right) + (\widehat{\mu}_1(x) - \widehat{\mu}_0(x)), & \text{for DR-learner.} \end{cases}
\tag{A.25}
$$

Note that, following the definition in (A.25), the following holds for both the IPW- and DR-learners:

$$
\pi(x) \, Y_\pi(x) + (1 - \pi(x)) \, Y_{1-\pi}(x) = Y(1) - Y(0).
\tag{A.26}
$$

Hence, applying (A.26) to the definition of the oracle score, we can write $V^*(x)$ as follows:

$$
\begin{aligned}
V^*(x) = |\widehat{\tau}(x) - (Y(1) - Y(0))| &= |\widehat{\tau}(x) - \pi(x) \, Y_\pi(x) - (1 - \pi(x)) \, Y_{1-\pi}(x)|, \\
&= |\pi(x) \, (\widehat{\tau}(x) - Y_\pi(x)) + (1 - \pi(x)) \, (\widehat{\tau}(x) - Y_{1-\pi}(x))|.
\end{aligned}
\tag{A.27}
$$

From the expressions in (A.24) and (A.27), we can see that $V_\varphi(x)$ is a mixture of the absolute values of the two random variables $(\widehat{\tau}(x) - Y_\pi(x))$ and $(\widehat{\tau}(x) - Y_{1-\pi}(x))$ with mixing proportions $\pi(x)$ and $1 - \pi(x)$, and $V^*(x)$ is the absolute value of the convex combination of the same random variables, $(\widehat{\tau}(x) - Y_\pi(x))$ and $(\widehat{\tau}(x) - Y_{1-\pi}(x))$, with weights $\pi(x)$ and $1 - \pi(x)$. Applying Lemma A2, it follows that $V_\varphi(x) \succeq_{mcx} V^*(x)$, $\forall x \in \mathcal{X}$, and from Lemma A1 we have that $V_\varphi \succeq_{mcx} V^*$. $\qquad \square$

**Stochastic orders for the signed distance conformity score:**

We have shown that for the absolute residual conformity score, $V_\varphi(\widehat{\tau}) = |\widehat{\tau}(X) - \widetilde{Y}_\varphi|$, the DR- and IPW-learners guarantee $V_\varphi \succeq_{mcx} V^*$ irrespective of the data distribution and underlying models. We now study the stochastic orders achieved by the signed distance conformity score proposed in [32], defined as: $V_\varphi(\widehat{\tau}) = \widehat{\tau}(X) - \widetilde{Y}_\varphi$. Let $u$ be a non-decreasing concave function, then we have:

$$
\begin{aligned}
\mathbb{E}[u(V_\varphi(\widehat{\tau}))] &= \mathbb{E}[u(\widehat{\tau}(X) - \widetilde{Y}_\varphi)] \\
&= \pi \, \mathbb{E}[u(\widehat{\tau}(X) - \widetilde{Y}_\pi)] + (1 - \pi) \, \mathbb{E}[u(\widehat{\tau}(X) - \widetilde{Y}_{1-\pi})] \\
&\leq \mathbb{E}[u(\pi \, \widehat{\tau}(X) - \pi \, \widetilde{Y}_\pi + (1 - \pi) \, \widehat{\tau}(X) - (1 - \pi) \, \widetilde{Y}_{1-\pi})] \\
&= \mathbb{E}[u(\widehat{\tau}(X) - (Y(1) - Y(0)))] = \mathbb{E}[u(V^*(\widehat{\tau}))],
\end{aligned}
\tag{A.28}
$$

where the inequality follows from the concavity of $u(.)$ and the last equality follows from (A.26). From Definition 1, (A.28) implies that $V^* \succeq_{(2)} V_\varphi$ (Table 2). Note, however, that the construction of predictive intervals with the signed error distance does not follow (5) since the signed distance score does not sort absolute errors from largest to smallest, but it sorts conformity scores from maximum positive error to maximum negative error. Hence, Theorem 1 does not apply to the signed distance error and the SOSD condition $V^* \succeq_{(2)} V_\varphi$ does not guarantee coverage.

Another more commonly used variant of the signed distance score was proposed in [25]. Given a quantile regression model $[\widehat{\tau}_l, \widehat{\tau}_h]$, [25] uses a variant of the signed error score defined as:

$$
V_\varphi(\widehat{\tau}) = \max\{\widehat{\tau}_l(X) - \widetilde{Y}_\varphi, \widetilde{Y}_\varphi - \widehat{\tau}_h(X)\}.
\tag{A.29}
$$

The construction of the predictive intervals in [25] follows (5). Note that $\max\{x, y\}$ can be written as:

$$\max\{x, y\} = \frac{x + y}{2} + \frac{|x - y|}{2}. \tag{A.30}$$

Applying (A.30) to (A.29), we can rewrite the conformity scores as:

$$V_\varphi(\widehat{\tau}) = \frac{\widehat{\tau}_l - \widehat{\tau}_h}{2} + \frac{|\widehat{\tau}_l + \widehat{\tau}_h - 2\widetilde{Y}_\varphi|}{2},$$
$$= \frac{\widehat{\tau}_l - \widehat{\tau}_h}{2} + \left| \frac{\widehat{\tau}_l + \widehat{\tau}_h}{2} - \widetilde{Y}_\varphi \right|. \tag{A.31}$$

Similarly, the oracle scores can be written as:

$$V^*(\widehat{\tau}) = \frac{\widehat{\tau}_l - \widehat{\tau}_h}{2} + \left| \frac{\widehat{\tau}_l + \widehat{\tau}_h}{2} - (Y(1) - Y(0)) \right|. \tag{A.32}$$

Let $\delta\tau_- = \frac{\widehat{\tau}_l - \widehat{\tau}_h}{2}$, $\delta\tau_+ = \frac{\widehat{\tau}_l + \widehat{\tau}_h}{2}$, and let $u$ be a non-decreasing convex function, then we have:

$$
\begin{aligned}
\mathbb{E}[u(V_\varphi(\widehat{\tau}))] &= \mathbb{E}\left[ u\left( \delta\tau_- + \left| \delta\tau_+ - \widetilde{Y}_\varphi \right| \right) \right] \\
&= \pi\, \mathbb{E}[u(\delta\tau_- + |\delta\tau_+ - \widetilde{Y}_\pi|)] + (1 - \pi)\, \mathbb{E}[u(\delta\tau_- + |\delta\tau_+ - \widetilde{Y}_{1-\pi}|)] \\
&\geq \mathbb{E}[u(\pi\, \delta\tau_- + \pi\, |\delta\tau_+ - \widetilde{Y}_\pi| + (1 - \pi)\, \delta\tau_- + (1 - \pi)\, |\delta\tau_+ - \widetilde{Y}_{1-\pi}|)] \\
&\geq \mathbb{E}[u(\delta\tau_- + |\delta\tau_+ - \pi\, \widetilde{Y}_\pi - (1 - \pi)\, \widetilde{Y}_{1-\pi}|)] \\
&= \mathbb{E}[u(\delta\tau_- + |\delta\tau_+ - (Y(1) - Y(0))|)] = \mathbb{E}[u(V^*(\widehat{\tau}))], \tag{A.33}
\end{aligned}
$$

where the first inequality follows from the convexity of $u$, the second inequality is an application of the triangle inequality, and the last equality follows from (A.26). Following Definition 2, (A.33) implies that $V_\varphi \succeq_{mcx} V^*$ when the base models apply quantile regression and the conformity scores follow the signed distance score in [25]. Hence, the coverage guarantee in Theorem 1 applies to conformal meta-learners based on conditional quantile regression.

## Appendix B: Literature Review

In this Section, we provide a comprehensive overview of the relevant literature. We categorize the previous literature into three distinct strands: (1) Bayesian methods for modeling individualized causal effects, (2) frequentist methods for inference on parameters pertaining to individualized causal effects, and (3) conformal methods for predictive inference of ITEs.

**(1) Bayesian methods.** Predictive inference on ITEs has been traditionally conducted using Bayesian methods. These methods place a prior distribution over the nuisance parameters $\mu_0$ and $\mu_1$, and then estimate the CATE function through the posterior distribution computed conditional on the observational dataset $\mathcal{D} = \{(X_i, W_i, Y_i)\}_i$. More formally, Bayesian procedures operate as follows:

$$Y(w) = \mu_w + \epsilon, \ \mu_0 \sim \Pi_0, \ \mu_1 \sim \Pi_1, \tag{A.34}$$

where $\Pi_0$ and $\Pi_1$ are priors over function spaces, and $\epsilon \sim \mathcal{N}(0, \sigma^2)$. Given the dataset $\mathcal{D}$, the posterior distributions over the two nuisance functions are used to estimate CATE as follows:

$$\widehat{\tau}(x) = \mathbb{E}_{\mu_1 \sim \Pi_1 | \mathcal{D}}[\mu_1(x)] - \mathbb{E}_{\mu_0 \sim \Pi_0 | \mathcal{D}}[\mu_0(x)]. \tag{A.35}$$

Furthermore, the posterior distributions $\Pi_0 | \mathcal{D}$ and $\Pi_1 | \mathcal{D}$ can be used to construct predictive intervals on ITEs through the posterior credible intervals of $P_{\Pi_0}(\mu_1(x) | \mathcal{D})$ and $P_{\Pi_1}(\mu_1(x) | \mathcal{D})$, i.e.,

$$P_{\mu_0, \mu_1 \sim \Pi_0, \Pi_1 | \mathcal{D}}(Y(1) - Y(0) \in \widehat{C}(X) \mid X = x) = 1 - \alpha. \tag{A.36}$$

Different incarnations of the Bayesian framework correspond to different choices of the prior $(\Pi_0, \Pi_1)$. Bayesian Additive Regression Trees (BART) [34] is one popular model for Bayesian nonparametric regression that was later adapted for causal effect estimation and inference [8, 35]. While [8] applies BART for estimation of CATE and inference of ITE out-of-the-box, [35] introduces new regularization techniques to incorporate different forms of inductive biases on the CATE target parameter. Another

popular choice of the prior $(\Pi_0, \Pi_1)$ is based on Gaussian processes where inductive biases are incorporated through different choices of the kernel function of a reproducible kernel Hilbert space [9].

Unlike the conformal framework, Bayesian methods do not provide coverage guarantees on their credible intervals. Frequentist coverage can be achieved in an asymptotic sense by Bayesian method under certain technical conditions [36]. Existing methods typically do not provide any finite-sample guarantees and the achieved empirical coverage depends on the choice of the prior. Models such as BART are typically applied with a default prior for all datasets, undermining their achieved coverage in diverse experimental setup. In [1], it was shown that Bayesian methods have a tendency to undercover ITEs in some data generation processes (e.g., in high-dimensional covariate spaces).

Bayesian methods also suffer from two key advantages compared to the conformal framework. First, Bayesian inference procedures and the computation of posterior distributions are typically tailored to specific model architectures. Generalizing exact Bayesian inference to arbitrary model architectures, including modern deep learning architectures, is not straightforward. Moreover, Bayesian inference is conducted either using computationally exhaustive Monte Carlo method (e.g., BART) or through expensive evaluation of an analytical posterior distribution that does not scale well with the number of training points (e.g., Gaussian process posteriors are $\mathcal{O}(n^3)$).

**(2) Frequentist methods.** Another model-specific approach to inference on causal parameters is the Causal Forest model in [7]. This procedure uses constructs pointwise confidence intervals on the CATE function $\tau(x)$ using conditional variance estimates based on the infinitesimal jackknife [37]. These intervals provide asymptotically valid coverage of CATE under mild regularity assumptions, but they provide no (asymptotic or finite-sample) guarantees on coverage for ITEs.

Another non-Bayesian approach for predictive inference of ITEs uses a quantile regression approach to train models that construct upper and lower bounds on the observed potential outcomes using a pinball loss [38]. While quantile regression can be repurposed for different model architectures, this approach does not provide finite-sample coverage guarantees and is empirically found to undercover outcomes, hence it is typically supplemented with a conformal prediction procedure [25].

**(3) Conformal methods.** The application of conformal prediction to ITE inference traces its origins back to the covariate shift problem. In [26], a variant of conformal prediction was proposed to address problem setups where training and testing data are drawn from different distributions, breaking the exchangeability assumptions required for valid inference.

In the treatment effect estimation setup, covariate shift arises because the distributions of the treatment groups differ from that of the target population. Recall from (8) that the conformity scores for the nuisance estimates $\widehat{\mu}_0$ and $\widehat{\mu}_1$ can be written as follows:

$$V_k^{(0)}(\widehat{\mu}_0) \triangleq V(X_k, Y_k(0); \widehat{\mu}_0), \forall k \in \mathcal{D}_{c,0}, \qquad V_k^{(1)}(\widehat{\mu}_1) \triangleq V(X_k, Y_k(1); \widehat{\mu}_1), \forall k \in \mathcal{D}_{c,1},$$

Both sets of conformity scores are sampled from the populations $P_{X|W=0}$ and $P_{X|W=1}$, respectively, and hence they are not exchangeable with the conformity scores sampled from target covariate distribution $P_X$. To retain the validity of intervals constructed using the sample quantiles of conformity scores, [26] introduces the notion of "weighted exchangeability" and proposes a weighted sample quantile with weights based on the likelihood ratios between training and testing distributions. [1] uses this method to construct valid intervals for potential outcomes and then proposes different approaches for combining these intervals to construct intervals for ITEs. Other work utilized the conformal prediction framework to conduct sensitivity analysis by devising hypothesis tests for the presence of hidden confounders [39, 40]. In [41, 42], conformal prediction has been utilized to address the closely related problem of off-policy evaluation. More recently, there has been further work developing methods for conformal prediction under covariate shift [43, 44, 45].

The key difference between our work and the aforementioned approaches is that we do not apply conformal prediction to the nuisance parameters, and instead conduct inference directly on the target parameter. We do so by applying conformalization with respect to transformed targets $\widetilde{Y}_\varphi$ for each data point in $\mathcal{D}$, and this way the covariate shift problem does not arise since $(X, \widetilde{Y}_\varphi)$ is sampled from the same distribution in calibration and testing data. One way to think about our approach as opposed to weighted conformal prediction (WCP) is that ours applies re-weighting to the outcomes whereas WCP re-weights the conformity scores. By moving the re-weighting step to the outcomes, we decouple the conformal procedure from the model, allowing for more flexible choice of inductive priors and accommodating models that do not estimate potential outcomes directly.

## Appendix C: Experimental Setup and Results

### C.1. Overview of the experimental setup

In all experiments, we used a Gradient Boosting model with 100 trees as the base model for nuisance estimation and quantile regression on pseudo-outcomes. We used the same base model in conformal meta-learners as well as the weighted CP (WCP) baselines to control for potential performance differences resulting from different choices of base models. Unless otherwise stated, all experiments followed a 90%/10% train/test split of each dataset, and each training set with further split into a 75%/25% proper training/calibration sets. All performance metrics (empirical coverage, expected interval width and RMSE) are evaluated on the testing data and averaged over 100 runs. The target coverage in all experiments was set $1 - \alpha = 0.9$.

We used the official implementations of the (naïve, exact and inexact) WCP baselines in the R package available at https://github.com/lihualei71/cfcausal. We executed these baselines through rpy2 (https://rpy2.github.io/) wrappers within our Python codebase.

### C.2. Details of the IHDP and NLSM datasets

In this Section, we provide details of the semi-synthetic datasets used in the experiments in Section 5.

**IHDP dataset.** The Infant Health and Development Program (IHDP) was a multi-site randomized controlled trial that studied the effect of early educational intervention on the cognitive development of low birth weight premature infants [46]. The trial targeted low-birth-weight, premature infants, and administered an intensive high-quality child care and home visits from a trained provider to the treatment group. In [8], a semi-synthetic benchmark was developed based on the covariate data in IHDP with simulated outcomes to assess the performance of different CATE estimation and ITE inference models when ground-truth counterfactual are known. The dataset comprised 25 covariates (6 continuous and 19 binary variables) capturing aspects related to children and their mothers. The benchmark dataset excluded a non-random subset of treated individuals [8]. The final dataset consists of 747 samples (139 treated and 608 control). The outcomes of all individuals were simulated whereas the treatment assignments followed the true assignments in the study.

The simulated outcomes model in IHDP followed simulation Setup B in [8]. In this Setup, the potential outcomes were simulated as follows: $Y(0) \sim \mathcal{N}(\exp((X + W)\beta), 1)$ and $Y(1) \sim \mathcal{N}(\exp(X\beta) - \omega, 1)$, where $W$ is an offset matrix, $\omega$ is set so that the ATE on the treated is always equal to 4, and the coefficients $\beta$ are sampled independently at random from $(0, 0.1, 0.2, 0.3, 0.4)$ with probabilities $(0.6, 0.1, 0.1, 0.1, 0.1)$. In our experiments, we used the 100 realization of the training and testing data released by [6] in https://www.fredjo.com/files/ihdp_npci_1-100.train.npz and https://www.fredjo.com/files/ihdp_npci_1-100.test.npz.

**NLSM dataset.** The National Study of Learning Mindsets (NLSM) is a large-scale randomized trial that studied the effect of a behavioral intervention, designed to instill students with a growth mindset, on the academic performance of students in secondary education in the US [3]. In [33], a semi-synthetic benchmark was developed based on the covariates of the NLSM study. Unlike IHDP, this benchmark does not use the real covariates of NLSM but rather buids a synthetic process to emulate NSLM in terms of the covariate distributions, data structures, and effect sizes. The final dataset comprised 10,000 data points. The data was generated based on 5 principles (See Section 2 and Appendix A in [33]). The principles are: (1) ATEs should be well-estimated from synthetic data by any reasonable procedure, (2) variability in CATE should be relatively modest, (3) treatment effect heterogeneity can be approximately recovered given complete knowledge of the correct functional form, (4) no additional unmeasured treatment effect moderation at the individual level, and (5) unexplained treatment effect heterogeneity at the group level should be present at reasonable levels.

### C.3. Further experiments using the ACIC dataset

In addition to the IHDP, NLSM and fully-synthetic experiments, we also conducted further experiments on 77 datasets from the 2016 Atlantic Causal Inference Competition (ACIC2016) [47]. We followed the processing steps in [16], creating a dataset of 4,802 data points and 55 covariates. Out of this base dataset, we created 770 simulation setups using the 77 settings proposed in [47]. The 77 settings represent different levels of complexity of response surfaces, varying degrees of confounding,

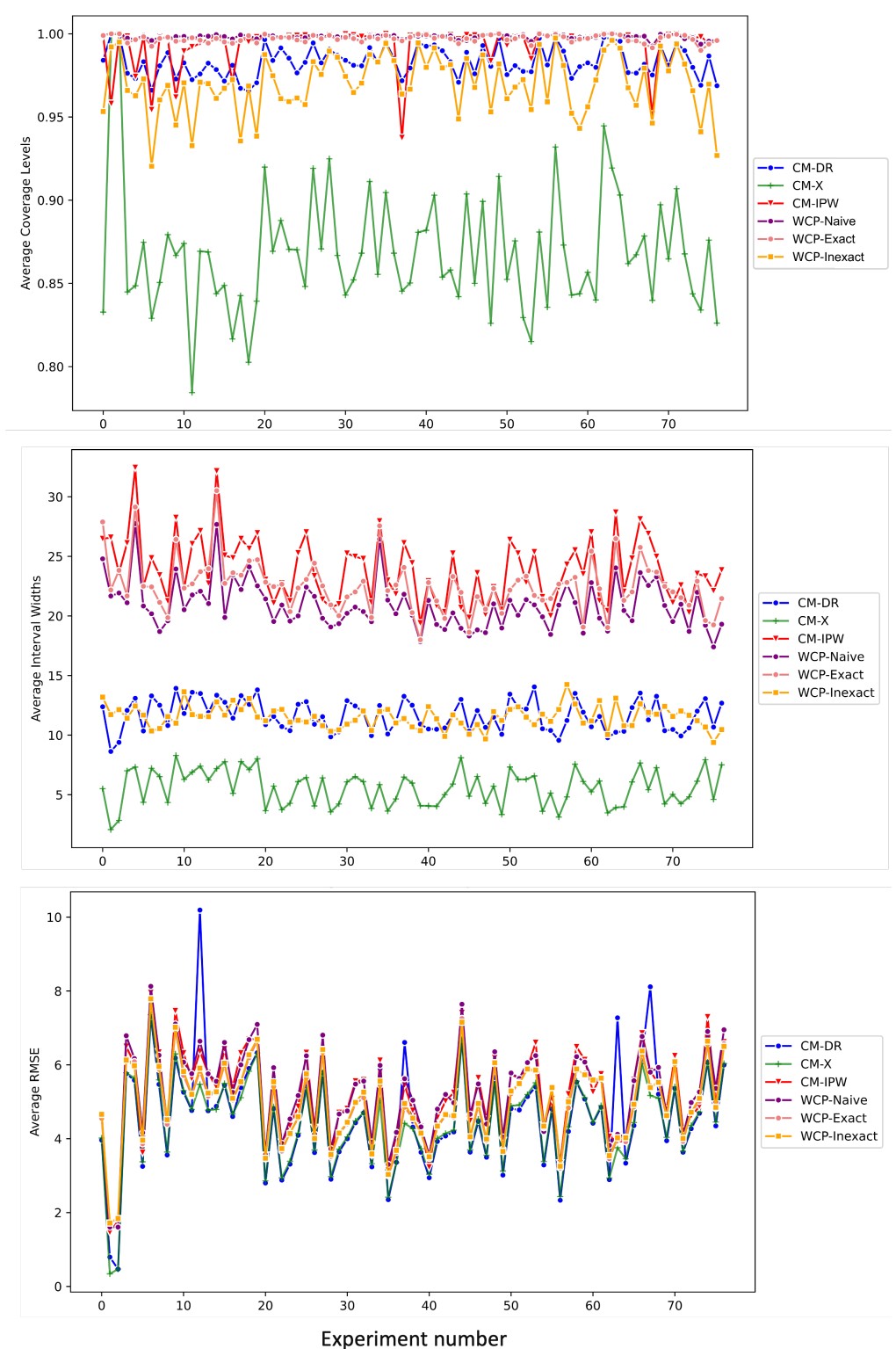

Figure C.1: Performance of all baseline across the 77 ACIC experiments.

overlap and effect heterogeneity. All settings use the same covariates but simlate different treatment assignments and response surfaces. We created 10 realizations of each of the 77 settings and average the performance of all baselines across the 10 runs.

The empirical coverage, average interval width and RMSE results across all 77 experiments are provided in Figure C.1. The comprehensive experiments in Figure C.1. align with the overall conclusions of the experiments in the main paper. For a target coverage of $1 - \alpha = 0.9$, all models achieved the target coverage with the exception of the X-learner. All models achieving the target coverage displayed conservative predictive intervals, acheiving coverage levels that exceed the 0.9 target. In terms of efficiency, the DR-learner and WCP-Inexact achieved comparable interval widths, significantly outperforming all other valid procedures (IPW-learner, WCP-exact and WCP-Naïve). Finally, the the DR-learner outperformed all valid procedure in terms of RMSE in almost all experiments, providing point estimation accuracy that is on par with the X-learner.

These results align with the overall conclusions of the experiments in the main paper. The DR-learner generally inherits the accurate point estimation of its underlying CATE meta-learner, provides a coverage guarantee for practically relevant values of target coverage and achieves this coverage with competitive efficiency making it a favorable approach for both point estimation of CATE and predictive inference of ITE.

