# OpenReview forum: "Conformal Meta-learners for Predictive Inference of Individual Treatment Effects"
_NeurIPS.cc/2023/Conference — NeurIPS 2023 oral_

### Official Review · Reviewer_SDYU · 2023-07-04

**Soundness:** 3 good
**Presentation:** 3 good
**Contribution:** 3 good
**Rating:** 6
**Confidence:** 4

**Summary:**

This paper develops a meta-learner-based approach for conformal inference of individual treatment effects (ITEs). Instead of predicting the missing outcome (the counterfactual), the proposed approach uses plug-in pseudo outcomes as the inference proxy. The validity of ITE inference is based on careful analysis of the statistical dominance of the ITE and the inference proxy. In this way, one no longer needs to adjust for the covariate shifts in applying conformal inference, and the validity holds in a model-free fashion (although the coverage guarantee is more limited). Overall, this paper finds a novel approach to predictive inference of ITEs, and the method works well in synthetic and real-world datasets.

**Strengths:**

1. Novel contribution to an important problem

This paper makes novel contributions, with a new methodology, to an important problem: uncertainty quantification and predictive inference of individual treatment effects. The proposed methodology significantly differs from the PO-prediction-based approaches in the literature, and the analysis contains interesting findings/perspectives. I believe these are important contributions to the literature of conformal inference, and might inspire more developments in the future.

2. Good writing quality and clarity

This paper is well-written and enjoyable to read. The challenges are clearly stated and the contributions are easy to capture. I still have some questions/suggestions for improving the presentation; please see my questions.

**Weaknesses:**

1. The analysis is somewhat limited to the known propensity case.

Since previous ITE methods already achieve validity in the known propensity case, the contribution of this paper is somewhat marginal in the sense that it does not push the limit of how well/ how much we can do for this problem (although i still appreciate the novel perspective provided here). However, the authors only mentioned the challenge of unknown propensity score at the end of the paper, which is a bit unsatisfactory.

2. The setting in synthetic datasets should be stated more clearly.

I think the most challenging part of ITE inference is when both POs are missing, and this is where the proposed approach becomes most interesting. However, when reading the experiments part I am not sure which situation we are in. Are we inferring the ITE for both POs missing case, and what is the calibration data here? I think further clarification will be helpful.

3. The benefits of this method in experiments is a bit unclear.

The experiment part contains rich information. However, I am not sure I fully understand in which cases the proposed method performs the best - is it particularly beneficial for both-PO-missing case, or one-PO-missing part? I thought in the latter case the previous approach should suffice, or it still can be improved due to no covariate shift adjustment? I also do not understand why IPW and DR lead to so long prediction intervals for the NLSM dataset. It is mentioned that "conformity scores have “very strong” dominance over oracle scores", but it is difficult to understand in which cases this might happen (strong signal? large noise? high nonlinearity?).

**Questions:**

1. What if the propensity score is unknown?

A natural question is whether the proposed method is robust to fitted propensities in the pseudo outcomes. I guess the dominance condition should still be (approximately) reasonable if the estimation is accurate (at least for DR?) since the expectation is the same, and the linear relation still holds? Currently the discussion on this point is a bit passive. But given the existing methods' good performance in this aspect, I was wondering whether more can be said about this problem.


2. Do we have a sense of how large $\alpha^*$ is for conditions (ii) and (iii) in Theorem 1, or practically used conformity scores?

It is also discussed that $\alpha^*$ is difficult to know, which I think is a hurdle to the current method. While it's said that $\alpha^*$ is evaluated in the experiments, I didn't understand how this can be done. Can we have some examples of the $\alpha^*$ under some parametric models + gaussian noise? Can we know how it changes with characteristics of the distributions?


3. A clarification question: Does it only apply to the setting without covariate shift?

From my understanding the method requires that the calibration data are exchangeable with the future point. This means for both-PO-missing parts we should also find such individuals in the calibration data, and similarly for single-PO-missing ones. If this is the case, it will be helpful to explicitly mention this because it would be helpful to practitioners.


4. Clarification in the presentation of the experiment part.

When reading the experiments part I am not sure which situation we are in. Are we inferring the ITE for both POs missing case, and what is the calibration data here? Did you evaluate the performance for the case with only one PO missing?


5. What affects the relative performance of meta-learner-based approach?

I see that the performance of the proposed approach is not always the best, and it can yield very long intervals for the NLSM dataset. It is important for practitioners to understand in which cases this method will perform well or poorly. Is it possible to give some guidance on how the performance changes with the data generating process, such as the coupling of POs, the strength of signal/noise?



---
Minor points & typos:

1. Should condition (ii) in Theorem 1 be $V_\varphi \succeq_{(2)} V^*$? Seems the current order is opposite to other types of conditions.

**Limitations:**

Yes.

---

> ### Author Rebuttal · Authors · 2023-08-08
>
> We would like to thank the reviewer for the thoughtful feedback and comments. Regarding condition (ii) in Theorem 1, the ordering is written correctly for the SOSD condition. Because these notations come from the decision theory literature, they are written to express risks associated with different distributions. SOSD is conventionally written in an opposite fashion to other types of conditions because it means that $V^*$ is "less risky" (i.e., has less spread) than $V_\varphi$.
>
> Please see below our point-by-point responses to your comments and questions.
>
> - **What if the propensity score is unknown?**
>
> When the data comes from an observational study with an unknown treatment mechanism, we need to estimate the propensity score $\pi$ and use this estimate in the pseudo-outcome transformation. In this case, the general conditions in Theorem 1 still hold, but Theorem 2 does not hold anymore. While an accurate estimate of $\pi$ will lead to a "approximate" validity, no formal result along these lines follow directly from Theorems 1 and 2.
>
> The good news is: to make progress in establishing validity of inference with estimated propensity scores, we can still use our stochastic ordering result in Theorem 1 combined with generalization bounds on estimated propensities to identify the conditions under which meta-learners with approximate propensities are valid. This is an interesting and natural extension of our work that can build on the novel stochastic dominance framework proposed in this paper to solve an even more challenging problem.
>
> While our analysis focuses on the case of known propensities, we believe that it still pushes the boundary of what we can do for the ITE inference problem, since all previous work applies CP to POs rather than ITE. To the best of our knowledge, this is the first paper that applies CP to infer ITE in an end-to-end fashion and enables the utilization of a broader range of meta-learners as the underlying CATE estimators. The novel stochastic ordering framework can also be used to solve more challenging versions of our setup, e.g., when hidden confounding exists or when propensity scores are unknown.
>
> - **How large is the validity region for conditions (ii) and (iii) in Theorem 1 or practically used conformity scores?**
>
> In almost all our experiments, we found that the pseudo-scores have FOSD over the oracle scores, which means that $\alpha^*=1$ and conformal meta-learners are valid for all target coverage. This is the case in all experiments in the main text (see Figures 4(c) and 5) as well as in the 77 experiments in the ACIC benchmark (provided in the Appendix). We believe that the IPW and DR may satisfy stronger stochastic orders (i.e., FOSD) than what we proved in Theorem 2. This is because our proof technique is based on an analysis of pointwise stochastic orders conditional on any feature point $X=x$. This is a sufficient but not necessary condition for the marginal scores to satisfy the MCX conditions in Theorem 2. It is likely that a stronger version of Theorem 2, showing that the meta-learner scores have FOSD over the oracle scores is viable.
>
> Characterizing $\alpha^*$ is a challenging problem because it requires inference on the crossing points of the CDFs of unknown distributions (i.e., the blue and red CDFs in Figures 4(c) and 5). This is a very interesting but technically-involved analysis that is of broad interest beyond the ITE inference problem, and would warrant a separate paper on its own.
>
> - **Both POs missing case vs. Only one PO missing**
>
> Please note that we only focus on the more challenging problem of inferring the ITE on a new point $X$ when both POs are missing. All our experiments and analyses are focused on this case. In fact, it is impossible to infer one PO with the IPW- and DR-learners because they do not provide explicit estimates of the POs in the first place. We will clarify this in the final manuscript.
>
> - **Does it only apply to the setting without covariate shift?**
>
> As mentioned in the previous point, we only focus on the case when both POs are missing. Please note that the key idea of our method is to apply a pseudo-outcome transformation to obtain a single dataset $(X_i, \widetilde{Y}_{\varphi, i})$, where $\widetilde{Y}_{\varphi}$ depends on the observables $X$, $Y$, and $W$. This transformation merges both the control and treated group into one dataset. Our framework applies the regression model and CP procedure to $(X_i, \widetilde{Y}_{\varphi, i})$. Since the covariates in this dataset are drawn from $P_X$, the tuple $(X_i, \widetilde{Y}_{\varphi, i})$ is exchangeable between calibration and test data, and there is no covariate shift between calibration and test pseudo-outcome labeled data points. Intuitively, for the IPW and DR-learners, the pseudo-outcome transformation adjusts for covariate shift in the output space rather than the covariate space.
>
> - **What affects the relative performance of meta-learner-based approaches?**
>
> We agree that a pre-fitting test for whether meta-learners would be a good approach for the problem at hand is important for practitioners to best use our method. It is hard to attribute performance to a single aspect of the data generation process (i.e., noise or non-linearity). Our hypothesis is that the “very strong dominance of pseudo-scores over oracle scores” (Line 339) that leads to the under-performance of meta-learners in terms of interval length is also associated with poor RMSE. We are working on a theory to establish the formal relationship between the meta-learner’s interval length and RMSE. If both quantities correlate, we can devise a hypothesis test for whether the RMSE of the conformal meta-learner exceeds that of the WCP methods, we can use this test to determine whether we should use conformal meta-learners for inference. Luckily, there are already plenty of formal model selection procedures for CATE estimation models that can be used for this purpose.

---

> > ### Comment · Reviewer_SDYU · 2023-08-15
> >
> > Thank you for your comments. While the answers to my first and last questions are not completely addressing them, I feel they are fair responses. I'll keep my scores as this paper makes valuable contributions yet still some assumptions/conditions remain not fully clear.
> >
> > One last question just out of curiosity: does this meta-learner idea apply to the confounded setting such as in Jin et al. 2023?

---

> > > ### Author Response · Authors · 2023-08-15
> > > **Thanks for your response**
> > >
> > > Thanks for your response. When hidden confounding exists, the causal effects are not identifiable so the coverage guarantees do not apply. However, one can operationalize the stochastic ordering framework to conduct sensitivity analysis for the meta-learner in a manner similar to Jin et al. 2023.

---

### Official Review · Reviewer_V6dY · 2023-07-07

**Soundness:** 3 good
**Presentation:** 4 excellent
**Contribution:** 3 good
**Rating:** 7
**Confidence:** 4

**Summary:**

The paper introduces a framework for inferring treatment effects, merging concepts of conformal prediction and meta-learner. This framework is characterized by its distribution-free validity and coverage guarantees. The authors utilize stochastic ordering techniques to substantiate the framework's validity.

**Strengths:**

The paper is well written and sounded. The innovative approach of employing stochastic ordering to validate the use of pseudo outcomes in conformal prediction is particularly noteworthy.

**Weaknesses:**

The concept of employing stochastic ordering is innovative; however, its practical application is impeded by the fact that the assumptions are inherently uncheckable

In comparison to the method proposed in [1], the current method exhibits better performance only in terms of the RMSE of the CATE. However, when considering coverage and average length, it does not surpass the performance of the inexact method.


[1] Lihua Lei and Emmanuel J Candès. Conformal inference of counterfactuals and individual treatment effects. Journal of the Royal Statistical Society Series B: Statistical Methodology, 83(5):911–938, 2021.

**Questions:**

(1). Assuming the propensity score is unknown, could we estimate it first and then follow the same procedure outlined in the paper?

(2). For the experimental results, it seems your method can be overly conservative.

(3). Figure 4(c) could be adjusted to resolve the issue of elements overlapping.

**Limitations:**

See weakness

---

> ### Author Rebuttal · Authors · 2023-08-08
>
> We would like to thank the reviewer for the thoughtful feedback and comments. In the final manuscript, we will fix the overlapping elements in Fig. 4(c).
>
> Please see below our point-by-point responses to your comments.
>
> - **Uncheckable assumptions involved in the stochastic ordering framework**
>
> We are unsure what assumptions the reviewer is referring to. The assumptions required for Theorems 1 and 2 in our paper to hold are:
>
> -- Exchangeability of training and testing points (Line 230). Note that this is a typical assumption that is guaranteed if the data points are drawn from i.i.d distributions.
>
> -- The propensity score is known. This is not an uncheckable assumption, and it holds in application domains where the treatment mechanism is known.
>
> Please note that **we do not assume stochastic ordering** between the pseudo-scores and oracle scores. Instead, we prove that if such ordering holds, then the resulting predictive intervals are valid (Theorem 1). In Theorem 2, we show that some of the well known pseudo-outcome transformations guarantee that stochastic ordering holds. Both results rely only on the two assumptions adobe and both assumptions are checkable.
>
> - **Performance comparison with other methods**
>
> The reviewer mentions that the proposed method only improves the RMSE of the CATE and not the interval width. Please note that the main goal of the paper is to develop an inference procedure that reuses some of the meta-learner architectures that perform well in terms of the RMSE while retaining validity of inference. That being said, the conformal DR-learner also outperforms all valid methods in terms of interval length (i.e., efficiency) in many experiments. In 77 experiments within the ACIC benchmark (provided in the Appendix), we found that DR-learners consistently outperformed all valid baselines (WCP exact and naive) in terms of efficiency and not just RMSE (Fig. c.1 in the Appendix).
>
> - **Plug-in estimates of the propensity scores**
>
> Yes, when the propensity scores are unknown, a plug-in estimate of $\pi$ can be used instead of the true propensities in the pseudo-outcome transformation. However, Theorem 2 will not hold in this case and a new analysis is needed. Fortunately, our proposed stochastic order framework can still be used to analyze the setting when propensity scores are approximate (please also refer to our responses to **Reviewers 855y** and **vDHA**).
>
> - **Conservatism of conformal meta-learners**
>
> We found the DR-learner intervals to be overly conservative only in the NLSM experiment. This is why we chose to highlight the NLSM experiment in the main text in order to demonstrate the cases when conformal meta-learners under-perform.
>
> Our empirical results do not show that its intervals are generally more conservative than other valid procedures. In most of our experiments (Fig. 4(b) and 4(c), Table 2 (IHDP)), the DR-learner intervals were comparable to the valid inference baselines (WCP exact and naive). In our extensive experiments on 77 datasets from the ACIC challenge (provided in the Appendix), we found that conformal DR-learners significantly outperforms all valid procedures based on WCP in terms of efficiency (Figure C.1 in the Appendix).
>
> In theory, there is no reason to believe that conformal meta-learners are destined to issue conservative intervals compared to the naive and exact WCP methods. These methods use Bonferroni correction or other multiple testing approaches to combine intervals on POs, which is also going to lead to conservative intervals. The conservatism of ITE intervals is a natural consequence of not observing counterfactuals.

---

> > ### Comment · Reviewer_V6dY · 2023-08-19
> >
> > I acknowledge that I have read the rebuttal and I have increased the rating.

---

### Official Review · Reviewer_vDHA · 2023-07-07

**Soundness:** 3 good
**Presentation:** 4 excellent
**Contribution:** 3 good
**Rating:** 7
**Confidence:** 4

**Summary:**

This paper adresses individual treatment effect (ITE) inference using conditional average treatment effect (CATE) meta learners.

The authors propose to wrap the meta-learners with a conformal prediction step so as to obtain valid confidence intervals for the estimated treatment effect. Since the CATE estimator induces a distribution shift wrt conformity scores, the coverage guarantee does not immediately translate into the desired coverage of actual ITE. In a who can do more can do less spirit, the authors point out that ITE coverage is ensured if conformity scores obtained from pseudo-outcomes stochastically dominate the scores obtained from actual « oracle » outcomes. The authors succeed at providing stochastic dominance results for several stochastic order / meta-learner pairs.



**Strengths:**

- the paper is very clear and pleasant to read
- the tackled problem is challenging and impactful
- the proposed solution is backup by both theoretical and experimental results

**Weaknesses:**

- the method is by-design over-conservative


**Questions:**

In my humble opinion, this is a very good paper. The problem is well formulated, the proposed solution is clearly exposed and easy to reproduce. The solution is theoretically proved to work (under some clearly stated conditions) which is confirmed by numerical studies.

Furthermore, the authors have correctly self-identified the limitations of their approach (knowledge of propensity score and hard to assess level range).

My only remark is that I would have like to see a synthetic experiment how far is the proposed solution to the optimum/oralce especially in terms of interval length.

**Limitations:**

- knowledge of the propensity score is required
- validity range of level $\alpha$ not always easy to determine
- over-conservatism of the prediction intervals

The two first points are already discussed by the authors in the submission.

---

> ### Author Rebuttal · Authors · 2023-08-08
>
> Thank you for your thoughtful comments and the feedback on our paper. Please see below some points that we would like to clarify in response to your comments.
>
> - **Knowledge of the propensity scores**
>
> Please note that this is not a unique limitation to our method. To date, all valid inference procedures for ITE in the literature require knowledge of the propensity scores. There are practical setups where this assumption will hold true, i.e., when the data comes from a randomized trial with or when treatment assignments follow a strictly known policy.  However, when the data comes from a real-world observational study, we need to estimate the propensity score $\pi$ and use this estimate in the pseudo-outcome transformation. In this case, the results in Theorem 2 will no longer hold and we will need to establish the conditions under which approximate propensity scores would give rise to valid inferences.
>
> On the positive side, the theoretical framework we proposed in this paper can help address this problem. Since the results of Theorem 1 still applies under approximate propensities, our stochastic ordering framework can still be used to analyze the validity of conformal meta-learners with approximate propensity scores. To approach this problem, one can use a PAC-style generalization bound on approximate propensities along with Theorem 1 to derive conditions for stochastic dominance (i.e. validity).
>
> - **Conservatism of conformal meta-learners**
>
> While we have not conducted a theoretical analysis of the efficiency of the DR-learner, our empirical results do not show that its intervals are more conservative than other valid procedures. In most of our experiments (Fig. 4(b) and 4(c), Table 2 (IHDP)), the DR-learner intervals are comparable to other valid procedures (WCP exact and naive) and are only outperformed by invalid procedures (inexact WCP). We chose to highlight the NLSM experiment in the main text in order to show the cases where conformal meta-learners under-perform, i.e., when the CDF of pseudo-scores and oracle scores deviate significantly (Figure 5). In our extensive experiments on 77 datasets from the ACIC challenge provided in the Appendix, we found that conformal DR-learners significantly outperforms all valid procedures based on WCP in terms of efficiency (Figure C.1 in the Appendix).
>
> In theory, there is no reason to believe that conformal meta-learners are destined to issue conservative intervals compared to the naive and exact WCP methods. These methods use Bonferroni correction or other multiple testing approaches to combine intervals on POs, which is also going to lead to conservative intervals. The conservatism of ITE intervals is a natural consequence of not observing counterfactuals—analyzing the optimal length of these intervals is an interesting direction for future work.
>
> - **Comparison with the oracle interval length in synthetic experiments**
>
> Thank you for this suggestion. We believe that we have already done this comparison in Figure 4(b) (blue vertical lines correspond to optimal interval widths). Please let us know if this addresses your question or if we misunderstood the oracle length you are referring to.

---

### Official Review · Reviewer_855y · 2023-07-31

**Soundness:** 3 good
**Presentation:** 2 fair
**Contribution:** 3 good
**Rating:** 7
**Confidence:** 3

**Summary:**

The paper deals with confidence intervals for the ITE prediction task, where the authors propose to use the framework of conformal prediction for meta learners. The learning of nuisance models for generating pseudo outcomes is done one separate data split, and the final regression models trained to predict pseudo outcomes are evaluated on a validation set to compute the empirical calibration distribution. The proposed approach for calibration removes the issues of covariate shift and inductive biases on nuisance models with the prior approaches of weighted conformal prediction. The authors also provide guarantees on when the confidence intervals for ITE with pseudo outcomes cover the confidence interval for ITE with true outcomes; and further validate their findings with synthetic and semi-synthetic datasets.

**Strengths:**

* ITE prediction is an extremely relevant problem in causal inference as population level effects (ATE, CATE) would not necessarily transfer to the individual level effects. Further, having an estimate of uncertainity along with ITE prediction is crucial for high-stakes application scenarios. Hence, the approach of conformal meta learners is highly relevant for the current challenges in causal inference.

* The proposed combination of conformal prediction with meta learners is novel to the best of my knowledge. Further, the ITE coverage results with pseudo outcomes for standard meta learners (DR Learner) are quite significant as it is non-trivial to understand when the predicted confidence intervals with observed information can represent the confidence intervals with counterfactual information.

* The proposed conformal meta learner approach is technically sound with theoretical proofs on their validity and empirical justification on a decently diverse set of benchmarks.

**Weaknesses:**

I do not think the paper has any serious weaknesses, but I have listed some of them in the questions section ahead. One major suggestion is that the paper could be written with better clarification for the section on conformal meta learner being robust to covariate shift and inductive biases. The arguments are stated informally and it is a bit hard to follow.

**Questions:**

* I do not understand the argument of authors that conformal prediction with meta learners is not affected by the choice of inductive biases of nuisance models. The pseudo outcomes are a function of the nuisance models, hence the predicted effect (ITE), which is learnt using pesudo outcomes, indeed depends indirectly on the nuisance models. Hence, the choice of nuisance models would affect the empirical calibration distribution, consequently the confidence intervals for ITE.

* I do not follow the argument of authors that conformal meta learners are not affected by the covariate shifts. The covariate shift between the control and treatment population should still affect the computation of pseudo outcomes with finite samples; hence the confidence intervals.

* Please clarify the assumption of knowing the true propensity model. What challenges would the authors in their analysis if they do not make this assumption.

**Limitations:**

Yes, the authors have addressed any potential negative societal impact of their work.

---

> ### Author Rebuttal · Authors · 2023-08-08
>
> We would like to thank the reviewer for the thoughtful and detailed feedback. Please see below our point-by-point responses to your comments.
>
> - **Conformal meta-learners and inductive biases**
>
> In the points below, we clarify our argument on how the proposed framework enables a more flexible set of choices of inductive priors for the CATE function.
>
> 1. First, we would like to stress that we *did not claim* that conformal meta-learners are *not affected by the choice of inductive priors* or that they are *robust to inductive biases*. In fact, inductive priors are not a problem that we need to overcome, but rather a design dimension that we need to consider when building our estimators. As we mention in Lines 146-147, covariate shift and inductive priors are *”two distinct characteristics of CATE estimation that interact with the conformal prediction procedure”*. As the reviewer rightly pointed out, the performance of the model (in terms of RMSE and efficiency) will depend on the choices for the models of $\mu_0$ and $\mu_1$. This is also evident in all our empirical results that compare the different meta-learners (i.e., different inductive biases) in Figures 4 and 5, as well as Tables 2 and 3.
>
> 2. The key finding of our paper is that a stochastic ordering framework can be used to evaluate the **validity (and not performance) of any meta-learner architecture** that encodes a specific inductive prior. This means that through the conformal meta-learners approach, we can apply CP to a broader class of models that correspond to a wider range of inductive priors. This enables us to conduct CP on top of models that typically perform well in terms of point estimation (RMSE), such as the DR-learner, instead of restricting to models that estimate and conformalize potential outcomes separately as in [1].
>
> 3. To sum up, our work does not propose a method that is *robust to inductive biases*, but proposes a framework for conducting valid inferences with a broader class of inductive priors than what was previously possible.
>
> 4. We believe that this confusion might have been caused by the wording in Line 47 which presents covariate shift and inductive biases as *two challenges*. We will improve the phrasing of this paragraph in the final version of the manuscript.
>
> - **Conformal meta-learners and covariate shift**
>
> Covariate shift arises in the treatment effect estimation problem because the covariates in the datasets $(X_i, Y_i(0))$ and $(X_j, Y_j(1))$ are drawn from distributions $P_{X|W=1}$ and $P_{X|W=0}$, respectively, which differ from the target distribution $P_X$.
>
> On the contrary, when we apply the pseudo-outcome transformation, we get a single dataset $(X_i, \widetilde{Y}_{\varphi, i})$, where $\widetilde{Y}_{\varphi}$ depends on the observables $X$, $Y$, and $W$. This transformation merges both the control and treated group into one dataset. Our framework applies the regression model and CP procedure to $(X_i, \widetilde{Y}_{\varphi, i})$. Since the covariates in this dataset are drawn from $P_X$, there is no covariate shift. Intuitively, for the IPW and DR-learners, the pseudo-outcome transformation adjusts for covariate shift in the output space rather than the covariate space.
>
> - **Knowledge of the propensity scores**
>
> To the best of our knowledge, all existing valid inference procedures for ITE require knowledge of the propensity scores. This assumption holds true when the data comes from a randomized trial with or when treatment assignments follow a strictly known policy. When the data comes from an observational study, we need to estimate the propensity score $\pi$ and use this estimate in the pseudo-outcome transformation. The challenge that would arise in this case is that the results on Theorem 2 will no longer hold and one will need to establish the conditions under which approximate propensity scores would give rise to valid inferences.
>
> Since the results of Theorem 1 still applies under approximate propensities, we note that our stochastic ordering framework can be useful in analyzing the validity of conformal meta-learners with approximate propensity scores. To approach this problem, one can use a PAC-style generalization bound on approximate propensities along with Theorem 1 to derive conditions for stochastic dominance (i.e. validity).
>
> - **Clarity of writing in Section 3**
>
> Thank you for this suggestion. In the final manuscript, we will improve the clarity of writing in Section 3 by clarifying how conformal meta-learners handle inductive biases and covariate shifts as explained above.

---

> > ### Comment · Reviewer_855y · 2023-08-16
> > **Good rebuttal!**
> >
> > Thanks for the good response during the rebuttal! My concerns are addressed and I have increased my score accordingly.

---

### Comment · Area_Chair_mT3T · 2023-08-18
**Reviewers, please respond to author's rebuttal.**

As a minimum, please acknowledge that you have read the rebuttal and whether it helps to change your rating, as the authors have tried to respond to your comments in the review. Thank you.

---

### Decision · Program_Chairs · 2023-09-21

**Decision:**

Accept (oral)

**Comment:**

This paper addresses an important and timely questions, namely to provide uncertainty quantification of individual treatment effect in causal inference. The reviewers agree that this is a well written paper and it is pleasant to read, that the tackled problem is challenging and impactful, and that the proposed solution is backup by both theoretical and experimental results. While some reviewers questioned the conservatism of the procedure, it is not showing in the simulation. The known propensity setting is also very common in the literature. It is determined that this paper should be accepted.